# Less is More: Federated Graph Learning with Alleviating Topology Heterogeneity from A Causal Perspective

**Lele Fu** [1]  **Bowen Deng** [1]  **Sheng Huang** [1]  **Tianchi Liao** [1]  **Shirui Pan** [2]  **Chuan Chen** [1]

## Abstract

Federated graph learning (FGL) aims to collaboratively train a global graph neural network (GNN) on multiple private graphs with preserving the local data privacy. Besides the common cases of data heterogeneity in conventional federated learning, FGL faces the unique challenge of topology heterogeneity. Most of existing FGL methods alleviate the negative impact of heterogeneity by introducing global signals. However, the manners of creating increments might not be effective and significantly increase the computation amount. In light of this, we propose the FedATH, an FGL method with Alleviating Topology Heterogeneity from a causal perspective. Inspired by the causal theory, we argue that not all edges in a topology are necessary for the training objective, less topology information might make more sense. With the aid of edge evaluator, the local graphs are divided into causal and biased subgraphs. A dual-GNN architecture is used to encode the two subgraphs into corresponding representations. Thus, the causal representations are drawn closer to the training objective while the biased representations are pulled away from it. Further, the Hilbert-Schmidt Independence Criterion is employed to strengthen the separability of the two subgraphs. Extensive experiments on six real-world graph datasets are conducted to demonstrate the superiority of the proposed FedATH over the compared approaches.

## 1. Introduction

Federated learning (FL) (Yang et al., 2019; Ye et al., 2023; Huang et al., 2024; Liao et al., 2024; Fu et al., 2025a; Hu et al., 2024) is a distributed model training approach that

[1]Sun Yat-sen University [2]Griffith University. Correspondence to: Chuan Chen <chenchuan@mail.sysu.edu.cn>.

*Proceedings of the 42nd International Conference on Machine Learning*, Vancouver, Canada. PMLR 267, 2025. Copyright 2025 by the author(s).

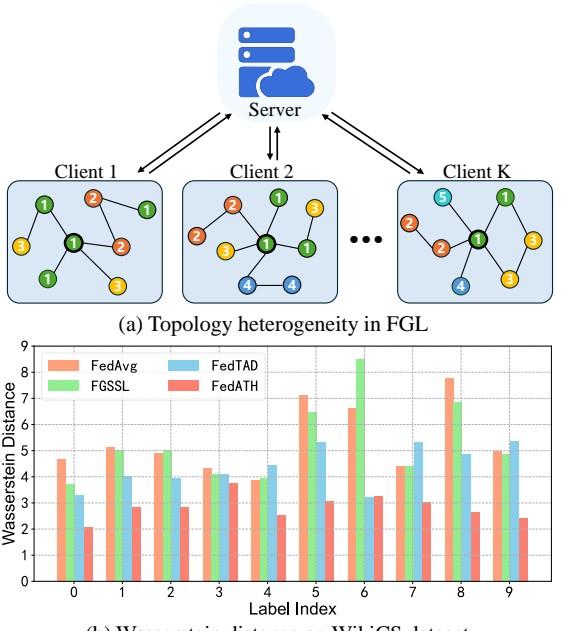

(a) Topology heterogeneity in FGL

(b) Wasserstein distance on WikiCS dataset

*Figure 1.* The illustration of topology heterogeneity in FGL. (a) shows that the nodes of same classes in different local graphs may be connected to these of diverse classes, resulting in the topology heterogeneity (the bolded node as an example). (b) compares the average wasserstein distance between the embedding from different local graphs for the same classes. It can be seen that the proposed FedATH effectively mitigates the embedding divergence caused by topology heterogeneity for different local graphs, thus alleviating the biased training of local GNNs.

has attracted wide attention for ensuring private data is not compromised. In light of this, various FL algorithms have flourished and been applied in many scenarios. Notably, an important assumption behind them cannot be ignored, that is, the samples on each client are independent and not correlated with each other, such as images and text. Meanwhile, graph data (Wu et al., 2020; Liu et al., 2022a; Wang et al., 2025; Deng et al., 2025; Cai et al., 2024b) is nowadays prevalent and may also present distributed storage such as transaction networks of multiple banks. Specifically, graph data has both feature attributes and topology structure, each node in a graph is connected with other nodes via

the edges. This complicated data format raises significant challenges for FL, which gives birth to federated graph learning (FGL) (Liu et al., 2024; Tan et al., 2023; Xie et al., 2021; Meng et al., 2024; Cai et al., 2024a; Wan et al., 2024).

As an emerging technique for distributed graph analysis, FGL aims to train a powerful global graph neural network (GNN) via incorporating multiple private graphs. Driven by practical requirements, many endeavors have been implemented. For example, (Zhang et al., 2021; Chen et al., 2024; Liu et al., 2022b; Tian et al., 2024) attempted to repair the neighbor nodes or global graph information, then allowing local GNNs to capture a wider scope of node information for enhancing the model training. The above approaches provide novel perspectives and achieve promising results. Unfortunately, FGL like traditional FL also suffers from the curse of data heterogeneity. The common manifestations of data heterogeneity are label shift and feature shift, both of which guide the local models to optimize in the direction of the locally optimal solutions, thereby weakening the ability of global model. In addition to the two kinds of data heterogeneity, a particular heterogeneity form in FGL needs to be emphasized, i.e., topology heterogeneity.

As presented in Fig. 1(a), topology heterogeneity shows that the nodes of the same classes may be connected to other nodes of different classes on different clients. When local GNNs encode the node information, the embedding composition of a node is not only determined by itself, but also influenced by its neighbor nodes. Then the topology heterogeneity across clients directly induces the heterogeneity of local embedding even for the same categories, resulting in biased training of local GNNs. Therefore, how to alleviate the impact of topology heterogeneity is a particular concern. Some efforts have been made for addressing this issue. For instance, Huang et al. (Huang et al., 2023a) used the global model to calibrate local embedding and structures. Zhu et al. (Zhu et al., 2024) generated a pseudo graph with the reliable knowledge from multiple clients, which served as the distillation data for training the global model. Xia et al. (Xia et al., 2024) augmented the local graph data via exploring the topological complementarity of various private graphs. In a nutshell, they attempt to conduct increments (e.g., additional global graphs or global representations) to shrink the heterogeneity of local graphs, but the used techniques, such as knowledge distillation and contrastive learning, significantly increase the computation and communication volume. More importantly, they might not be effective in reducing the negative impact of topology heterogeneity.

Fig. 1(b) compares the average wasserstein distance (WD) between the embedding from different local graphs for the same classes, where the WD is used to measure the similarity between distributions. Smaller WD indicates that the two distributions are more similar, and the opposite is

less similar. It can be seen that existing SOTA FGL methods such as FGSSL (Huang et al., 2023a) and FedTAD (Zhu et al., 2024) cannot consistently guarantee that the embedding divergences from different clients are decreased, demonstrating that they fail to effectively address the problem of topology heterogeneity. As a result, we reflect on whether conducting increments is really conducive to mitigating topology heterogeneity. Are there better ways to accomplish this goal? Inspired by the causal theory, we believe that not all edges in a local topology are necessary for the training objective, and only a part of them plays a deterministic role, while the rest is unimportant or even has a negative effect. Less local topology information may make more sense. The component consisting of deterministic edges is considered as the causal subgraph, while the rest is considered as the biased subgraph. When each client explores the causal subgraph, the node embedding is as relevant as possible to the training objective, thereby mitigating topology heterogeneity and preventing biased training of local GNNs.

In this paper, we propose an FGL method with Alleviating Topology Heterogeneity (FedATH) across multiple clients. Concretely, we adopt an edge evaluator to assess the importance of each edge. Based on the assessment results, the local graphs are divided into causal subgraphs and biased subgraphs. Further, a dual-GNN architecture is used to encode the two kinds of subgraphs into corresponding representations. The cross entropy loss is used to reinforce the correlation of causal representation with the training objective while the negative entropy loss is used to disassociate the biased representation from the training objective. To enhance the separability of them, the Hilbert-Schmidt Independence Criterion (HSIC) is introduced to maximize their independence. Notably, only the local causal GNNs are uploaded to the server for aggregation without increasing communication burden. Fig. 1(b) shows that the proposed FedATH significantly reduces the embedding divergence between varying local graphs, this is because the exploration of causal subgraphs effectively handles the topology heterogeneity across multiple clients. Generally, the principal contributions of this paper are concluded as follows:

- We provide a novel perspective on the problem of topology heterogeneity in FGL, mitigating the negative impact of topology heterogeneity across different clients by diminishing superfluous information rather than creating new increments.

- Inspired by the causal theory, we divide the local graphs into causal and biased subgraphs with the aid of the edge evaluator, and the HSIC is adopted to enforce their separability. Finally, only the local causal GNNs are shared for aggregation.

- A large number of experiments are conducted on six

real-world graph datasets, the experimental results demonstrate that the proposed FedATH is more superior than the SOTA conventional FL and FGL methods.

## 2. Related Work

### 2.1. Federated Learning

With the increased awareness of protection for private data, FL has flourished as a means of distributed model training. As the pioneering algorithm, FedAvg (McMahan et al., 2017) has demonstrated the superiority of FL, but it is highly sensitive to the heterogeneous data. Therefore, how to overcome the detrimental effects induced by heterogeneous data has always been a central concern in FL. For the scenario of heterogeneous labels, (Li et al., 2020; Karimireddy et al., 2020; Li et al., 2021; Fu et al., 2025c; Huang et al., 2025) corrected the bias between the local models and global model to prevent the local models from falling into local optima. Knowledge distillation is an effective method for transferring information between different models, which is also introduced into FL to address the issue of catastrophic forgetting caused by heterogeneous data. (Li & Wang, 2019; Zhu et al., 2021; Shao et al., 2024; Xie et al., 2024) passed the global knowledge to clients by leveraging knowledge distillation, promoting the generalization of local models. For the scenario of heterogeneous features, (Hong et al., 2023; Wang et al., 2023; Zhang et al., 2024) adopted an adversarial training method, eliminating the divergence between various domains by fooling the discriminator. (Huang et al., 2023b; Li et al., 2023a; Yan et al., 2024; Qi et al., 2023; Meng et al., 2025; Fu et al., 2025b) explored the generalized global prototypes and aligned the representation spaces of different clients through contrastive learning. Despite the aforementioned methods achieve impressive results, they perform unsatisfactorily when the clients' private data is graph-structured, which is because graph data is far more complex than common image or text data. Accordingly, it is necessary to develop tailored FGL algorithms.

### 2.2. Federated Graph Learning

FGL aims to train a decent GNN with distributed graph data. Unlike traditional FL strategies, FGL requires to additionally consider the impact of topology structure on model training. Overall, FGL is categorized into two types based on the graph data format on the clients. The first type is the graph-level, where each graph is considered as a sample such as molecular graphs and protein graphs. The second type is the node-level, where each node in a graph serves as a sample such as citation network and social network. For the graph-level, each client has a set of graphs. Xie et al. (Xie et al., 2021) dynamically grouped clients into different clusters according to the gradients of local GNNs.

Tan et al. (Tan et al., 2023) proposed to share the topology encoding networks while maintaining the feature encoding networks specific. Pan et al. (Pan et al., 2024) devised an incentive mechanism to retain the fairness among multiple agents in federated graph system. For the node-level, each client stores a subgraph. Zhang et al. (Zhang et al., 2021) generated the missing neighbor nodes for each client with federated training, promoting the performance of local GNNs. Chen et al. (Chen et al., 2021) developed a graph sampling strategy and federated graph convolutional operation for distributed graph data. Li et al. (Li et al., 2023b) considered the impact of topology structure and proposed a topology-aware federated aggregation manner. Kong et al. (Kong et al., 2024) adopted a federated fusion strategy for local anomalous neighbor embedding, enhancing the difference between anomalous nodes and neighbor nodes.

## 3. Preliminaries

**Graph Neural Networks.** Given a graph dataset $G = (V, E, \mathbf{X})$, $V$ denotes the node set, $E$ denotes the edge set, and $\mathbf{X} \in \mathbb{R}^{N \times d}$ denotes the node feature matrix, where $N$ and $d$ are the number of nodes and feature dimension, respectively. For each node $v_i \in V$, it has a feature attribute $\mathbf{x}_i$ (the $i$-th row of $\mathbf{X}$) with label $y_i \in [C]$, where $C$ is the number of categories. GNNs aim to aggregate the neighborhood information of nodes to improve the discriminability of their representations through a certain defined information propagation mechanism. Generally, the calculation of the $l$-layer GNN is formulated as

$$\mathbf{h}_i^{l+1} = \delta(\mathbf{h}_i^l, \text{AGG}(\mathbf{h}_j^l, e_{ij}) | \forall j \in V), \quad (1)$$

where $\mathbf{h}_i^l$ is the embedding of the $l$-layer for the $i$-th node, $e_{ij}$ denotes the edge between the $i$-th and $j$-th nodes, $\text{AGG}(\cdot)$ is the defined aggregation operator of neighbor nodes, $\delta(\cdot)$ denotes the activation function. Especially, $\mathbf{h}_i^0 = \mathbf{x}_i$ is the raw feature.

**Federated Graph Learning.** In a federated graph system, a centralized server and $K$ clients are included, each client stores a private graph dataset $G_k = (V_k, E_k, \mathbf{X}_k)$. Each node $v_k^i$ is characterized as $(\mathbf{x}_k^i, y_k^i | \forall i \in [N_k])$, where $N_k$ is the number of the $k$-th local graph's nodes. The vanilla FGL is to directly transplant FedAvg into the federated scenario. Thus, the objective optimization of FGL is written as

$$\min_{\mathbf{W}_1, \dots, \mathbf{W}_K} \sum_{k=1}^{K} \frac{1}{K} R_k(\mathbf{W}_k) \quad (2)$$

where $\mathbf{W}_k$ is the parameter of the $k$-th local GNN, $R_k$ denotes the $k$-th empirical risk and is defined as

$$R_k(\mathbf{W}_k) = \mathbb{E}_{(\mathbf{x}_k^i, y_k^i)}(F_k(\mathbf{W}_k | (\mathbf{x}_k^i, y_k^i))), \quad (3)$$

where $F_k(\cdot)$ denotes the loss function for the $k$-th client. When each communication round completes, the updated

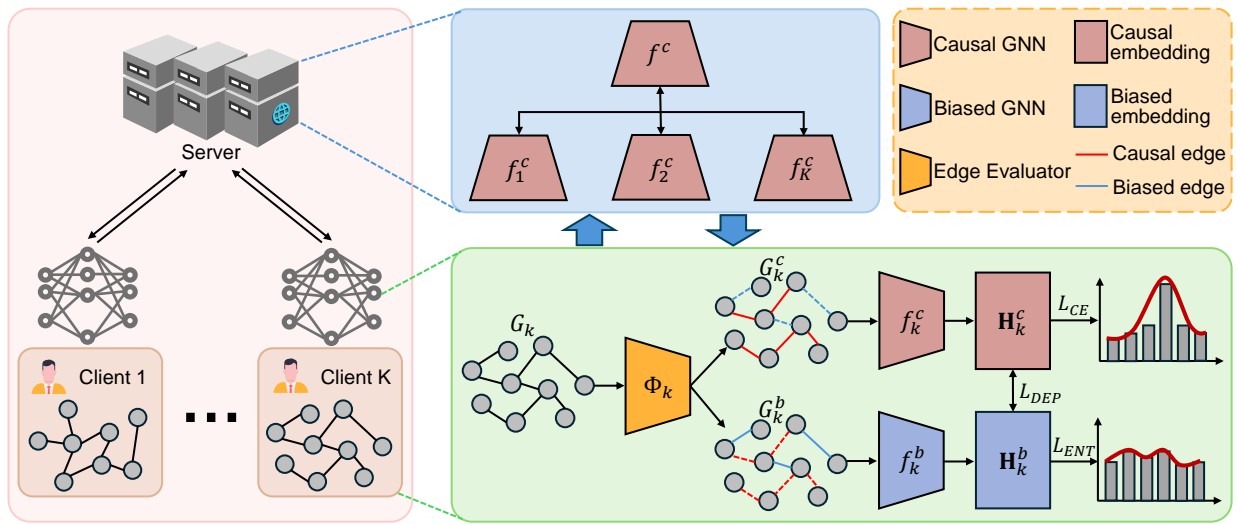

*Figure 2.* The overview of the proposed FedATH. The pink box presents the general FGL framework, the green box shows the local training process on each client, and the blue box shows the aggregation procedure on the server. In particular, the yellow box displays the meaning of used graphics.

global GNN is obtained by

$$\mathbf{W} = \sum_{k=1}^{K} \frac{N_k}{N} \mathbf{W}_k, \tag{4}$$

where $N = \sum_{k=1}^{K} N_k$ is the total number of nodes. However, simple aggregation inescapably causes the performance degeneration due to the topology heterogeneity across multiple clients.

## 4. Proposed Method

Concretely, the proposed FedATH consists of two important modules: Subgraph Division via Edge Evaluation and Disentanglement of Causal and Biased Representations. Fig. 2 illustrates the framework of the proposed FedATH. The details are elaborated as follows.

### 4.1. Inspiration from Causal Theory

The proposed FedATH is inspired by the causal theory. To better understand it, the graph generation procedure from the structure causal model (SCM) (Schölkopf et al., 2021; Fan et al., 2022) has to be first elaborated. As shown in Fig. 3(a), there are four kinds of causal relationships between various variables in the general cases. (1) $C \rightarrow G \leftarrow B$. The observed graph is produced by the unobserved causal variable $C$ and biased variable $B$. (2) $C \rightarrow Y$. The causal variable $C$ fundamentally determines the semantics $Y$. (3) $C \leftrightarrow B$. There may be redundant entanglement between $C$ and $B$. (4) $G \rightarrow R \rightarrow Y$. Most GNNs directly map the raw graph $G$ into latent representation $R$, then yields the semantic label $Y$.

In FGL, the nodes of same categories in different local graphs might be connected to ones of various categories due to the heterogeneous topology, contributing to the biased training of local GNNs. However, inspired by the above SCM, not all edges in a topology are essential for the node semantics, some are the biased factors and even play the negative roles. If each client adopts the common GNN encoding manner, the representation is mixed with causal and biased variables, which is not conducive to the homogeneity across clients. Hence, we expect to identify which edges are determinant for the semantic and explore the key topology on each client. *The part formed by keeping the important edges are regarded as causal subgraph, while the rest are regarded as biased subgraph.* When each client disentangles the local graph into causal and biased subgraphs, the labels are only associated with the causal subgraph and stripped from the biased subgraph, then the topology heterogeneity of various local graphs can be alleviated.

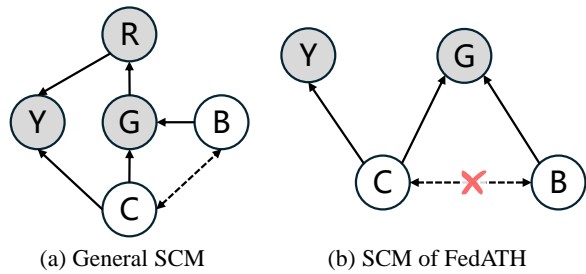

(a) General SCM      (b) SCM of FedATH

*Figure 3.* The SCMs of common FGL methods and the proposed FedATH, where the grey circle denotes the observed variable and the white circle denotes the latent variable.

## 4.2. Subgraph Division via Edge Evaluation

To divide the local graph into causal and biased subgraphs, we propose to train an edge evaluator $\Phi_k$ on each client for deciding the affiliation of edges. Specifically, for the $k$-th local graph $G_k = (V_k, E_k, \mathbf{X}_k)$, the contribution degree of each edge $e_{ij} \in E_k$ between node $v_i \in V_k$ and node $v_j \in V_k$ is evaluated by

$$c_{ij} = \Phi_k([\mathbf{x}_k^i || \mathbf{x}_k^j]), \tag{5}$$

where $||$ denotes the concatenation of two vectors, the evaluator $\Phi_k$ can be specified as a Multi-Layer Perceptron (MLP). Further, to enable $c_{ij}$ to be probabilistically meaningful, it is mapped into $[0, 1]$ by

$$\omega_{ij} = \mathrm{sigmoid}(c_{ij}), \tag{6}$$

where $\omega_{ij}$ can be viewed as the probability that edge $e_{ij}$ belongs to the causal subgraph, while $1 - \omega_{ij}$ is the probability that edge $e_{ij}$ belongs to the biased subgraph. Frequently, the adjacency matrix $\mathbf{A}_k \in \mathbb{R}^{N_k \times N_k}$ of a local graph $G_k$ is conducted based on the edge set $E_k$ for an intuitive expression. Likewise, the edge mask matrices of causal and biased subgraphs can be formulated as $\boldsymbol{\Omega}_k^c = [\omega_{ij}] \in \mathbb{R}^{N_k \times N_k}$ and $\boldsymbol{\Omega}_k^b = [1 - \omega_{ij}] \in \mathbb{R}^{N_k \times N_k}$, respectively. Further, the adjacency matrix $\mathbf{A}_k$ is disentangled into causal adjacency matrix $\mathbf{A}_k^c = \mathbf{A}_k \odot \boldsymbol{\Omega}_k^c$ and biased adjacency matrix $\mathbf{A}_k^b = \mathbf{A}_k \odot \boldsymbol{\Omega}_k^b$, where $\odot$ denotes the Hadamard product. When $\mathbf{A}_k^c$ and $\mathbf{A}_k^b$ are obtained, the causal subgraph $G_k^c = (\mathbf{A}_k^c, \mathbf{X}_k)$ and the biased subgraph $G_k^b = (\mathbf{A}_k^b, \mathbf{X}_k)$ can be constructed.

With the help of the edge evaluator, each client explores the causal and biased subgraphs, which further assist the GNNs to explore the corresponding latent representations. Notably, the edge evaluator is only trained on local clients and not shared, which elicits two advantages. First, federated learning emphasizes efficient communication, while a private evaluator does not incur additional communication burdens. Second, the topologies of different local graphs have different characteristics, a private edge evaluator captures the personalized information, and a shared evaluator may interfere with the extraction of local causal and biased subgraphs. Moreover, the experimental comparison between shared and unshared edge evaluator is expanded in detail in the experimental section.

## 4.3. Disentanglement of Causal and Biased Representations

For causal subgraph $G_k^c$ and biased subgraph $G_k^b$, how to guarantee their disengagement becomes the principal concern. In response to this problem, we propose to adopt a dual-GNN architecture to encode the two subgraphs into latent representations and achieve the disengagement of the two that is driven by the designed losses upon them.

Concretely, a causal GNN $f_k^c$ and a biased GNN $f_k^b$ are equipped on each client, they encode the corresponding subgraphs into latent space as

$$\begin{aligned} \mathbf{H}_k^c &= f_k^c(G_k^c | \mathbf{W}_k^c) \\ \mathbf{H}_k^b &= f_k^b(G_k^b | \mathbf{W}_k^b), \end{aligned} \tag{7}$$

where $\mathbf{W}_k^c$ and $\mathbf{W}_k^b$ are the parameters of $f_k^c$ and $f_k^b$, respectively. For a concise denotation, the subscript $k$ is omitted in the following presentation. The causal representation $\mathbf{H}^c$ is considered to capture the information that is strongly relevant to the objective, the cross entropy is used to constrain its distance from the ground truth labels, which is written as

$$\mathcal{L}_{CE} = -\mathbb{E}_{(\mathbf{h}_i^c, \mathbf{y}_i | \forall i \in [N_t])} \sum_{i=1}^{N_t} \mathbb{1}_{\mathbf{y}_i} \log(\mathrm{softmax}(\mathbf{h}_i^c)), \tag{8}$$

where $\mathbf{h}_i^c$ is the $i$-th row of $\mathbf{H}^c$, $\mathbb{1}_{\mathbf{y}_i}$ denotes the one-hot vector for the label $\mathbf{y}_i$, $N_t$ denotes the number of training nodes. Due to the absorption of unnecessary or even negative information from neighbor nodes, the biased representation $\mathbf{H}^b$ is regarded to be not discriminative, as a direct result of which its predicted label probability distribution appears smooth rather than sharp. Hence, the negative entropy loss is adopted to achieve the goal:

$$\mathcal{L}_{ENT} = -\mathbb{E}_{(\mathbf{h}_i^b | \forall i \in [N_k])} \sum_{i=1}^{N_k} \log(\mathrm{softmax}(\mathbf{h}_i^b)), \tag{9}$$

where $\mathbf{h}_i^b$ is the $i$-th row of $\mathbf{H}^b$, $N_k$ is the number of nodes in the $k$-th local graph.

According to the causality presented in Fig. 3, the causal representation $\mathbf{H}^c$ and the biased representation $\mathbf{H}^b$ should be independent of each other. To measure the dependence between $\mathbf{H}^c$ and $\mathbf{H}^b$, the Hilbert-Schmidt Independence Criterion (HSIC) is introduced. Give two variables $\mathbf{P} = [\mathbf{p}_1, \mathbf{p}_2, ..., \mathbf{p}_N]$ and $\mathbf{Q} = [\mathbf{q}_1, \mathbf{q}_2, ..., \mathbf{q}_N]$, the mapping functions $\psi$ and $\phi$ map them into kernel spaces $\mathcal{P}$ and $\mathcal{Q}$: $\psi(\mathbf{p}) \in \mathcal{P}, \phi(\mathbf{q}) \in \mathcal{Q}$. Then, the inner product for two vectors in kernel spaces can be written as $\kappa_1(\mathbf{p}_1, \mathbf{p}_2) = \langle \psi(\mathbf{p}_1), \psi(\mathbf{p}_2) \rangle$, $\kappa_2(\mathbf{q}_1, \mathbf{q}_2) = \langle \phi(\mathbf{q}_1), \phi(\mathbf{q}_2) \rangle$. Then, it has following definition.

**Definition 4.1.** Given a set of independent observed variables $\mathcal{X} := \{(\mathbf{p}_1, \mathbf{q}_1), ..., (\mathbf{p}_N, \mathbf{q}_N)\}$, an empirical estimator of $\mathrm{HSIC}(\mathcal{X}, \mathcal{P}, \mathcal{Q})$ is defined as

$$\mathrm{HSIC}(\mathcal{X}, \mathcal{P}, \mathcal{Q}) = (N - 1)^{-2} \mathrm{Tr}(\mathbf{K}_1 \mathbf{C} \mathbf{K}_2 \mathbf{C}), \tag{10}$$

where $Tr(\cdot)$ denotes the trace of a matrix. $\mathbf{K}_1$ and $\mathbf{K}_2$ are the Gram matrices, whose each entry is computed by $\mathbf{K}_{1,ij} = \kappa_1(\mathbf{p}_1, \mathbf{p}_2)$, $\mathbf{K}_{2,ij} = \kappa_2(\mathbf{q}_1, \mathbf{q}_2)$ respectively. $\mathbf{C}$ is the centralized matrix for the Gram matrix and is defined as $\mathbf{H} = \mathbf{I} - 1/N$, where $\mathbf{I} \in \mathbb{R}^{N \times N}$ is an identity matrix.

Therefore, the dependence loss between $\mathbf{H}^c$ and $\mathbf{H}^b$ can be written as

$$\mathcal{L}_{DEP} = \text{HSIC}(\mathbf{H}^c, \mathbf{H}^b) = (N-1)^2 \, \text{Tr}(\mathbf{K}_1 \mathbf{C} \mathbf{K}_2 \mathbf{C}) \quad (11)$$

where the inner kernels are specified as $\mathbf{K}_1 = \mathbf{H}^c \mathbf{H}^{c^T}$, $\mathbf{K}_2 = \mathbf{H}^b \mathbf{H}^{b^T}$. Through minimizing the objective expressed as Eq. (11), the disentanglement of causal and biased representations are enhanced. Here, the overall loss function is formulated as

$$\mathcal{L} = \mathcal{L}_{CE} + \mathcal{L}_{ENT} + \lambda \mathcal{L}_{DEP}, \quad (12)$$

where $\lambda$ is the trade-off parameter. When the local training completes, only the causal GNN is uploaded to the server for aggregation while the biased GNN and the edge evaluator remain private.

## 5. Theoretical Analysis

Here, we provide the generalization analysis for the proposed FedATH. Denote $\mathcal{D}$ and $\mathcal{D}_k$ the global and local distributions, respectively. $\tilde{\mathcal{D}}_k$ denotes the empirical local distribution. $h_k$ is the local hypothesis learned on the local empirical distribution $\tilde{\mathcal{D}}_k$ and defined as $h_k : \mathcal{X} \to \mathcal{Y}$, mapping the data features into predicted labels. $h = 1/K \sum_{k=1}^{K} h_k$ denotes the global hypothesis integrated by local hypothesis. $\mathcal{H}$ denotes the hypothesis space of VC-dimension $d$. Moreover, without losing generality, it is specified that $N_1 = ... = N_K = m$.

**Theorem 5.1.** *Given an FGL system with global distribution $\mathcal{D}$ and local distribution $\mathcal{D}_k$, with the the probability at least $1 - \delta$ $(0 < \delta \leq 1)$, the generalization error for any hypothesis $h_k$ satisfies*

$$\mathcal{R}_{\mathcal{D}}(h) \leq \frac{1}{K} \sum_{k \in [K]} \hat{\mathcal{R}}_{\tilde{\mathcal{D}}_k}(h_k)$$

$$+ \frac{1}{K(K-1)} \sum_{k \in [K]} \sum_{l \neq k}^{K} d_{\mathcal{H} \Delta \mathcal{H}} \left( \tilde{\mathcal{D}}_k, \tilde{\mathcal{D}}_l \right) + \epsilon$$

$$+ \frac{1}{K} \sum_{k \in [K]} \lambda_k + \sqrt{\frac{4}{m} \left( d \log \frac{2em}{d} + \log \frac{4K}{\delta} \right)}, \quad (13)$$

where $\hat{\mathcal{R}}_{\tilde{\mathcal{D}}_k}(h_k)$ denotes the empirical risk on $\tilde{\mathcal{D}}_k$, $d_{\mathcal{H} \Delta \mathcal{H}}(\tilde{\mathcal{D}}_k, \tilde{\mathcal{D}}_l)$ is the $\mathcal{H}$-distance between $\tilde{\mathcal{D}}_k$ and $\tilde{\mathcal{D}}_l$, $\epsilon$ denotes a upper-bound constant with respect to $d_{\mathcal{H} \Delta \mathcal{H}}(\tilde{\mathcal{D}}_l, \tilde{\mathcal{D}}), \forall l \in [K]$, $\lambda_k = \min_h(\mathcal{R}_{\mathcal{D}}(h) + \mathcal{R}_{\mathcal{D}_k}(h))$ denotes the optimal risk on $\mathcal{D}$ and $\mathcal{D}_k$.

**Theorem 5.1** reveals that the generalization error of an FGL system depends mainly on two factors: the distribution divergence between local graph data $d_{\mathcal{H} \Delta \mathcal{H}}(\tilde{\mathcal{D}}_k, \tilde{\mathcal{D}}_l)$ and the number of observed samples $m$. In particular, we further have following corollary for $d_{\mathcal{H} \Delta \mathcal{H}}(\tilde{\mathcal{D}}_k, \tilde{\mathcal{D}}_l)$.

**Corollary 5.2.** *Given an FGL system, the $k$-th and $l$-th empirical local distributions are denoted as $\hat{\mathcal{D}}_k$ and $\hat{\mathcal{D}}_l$, its generalization error follows **Theorem 5.1**. For $d_{\mathcal{H} \Delta \mathcal{H}}(\tilde{\mathcal{D}}_k, \tilde{\mathcal{D}}_l)$, the following inequality holds.*

$$d_{\mathcal{H} \Delta \mathcal{H}}(\tilde{\mathcal{D}}_k, \tilde{\mathcal{D}}_l) \leq 1 + \sup_f B_W^2 B_X \left( \frac{1}{D_{\min} m} + \frac{1}{D_{\min}^2} \right)$$

$$(\|\mathbf{A}_k\|_F + \|\mathbf{A}_l\|_F) + \frac{B_W^2}{m} \|\mathbf{X}_k - \mathbf{X}_l\|_F, \quad (14)$$

*where* $\sup$ *denotes the supremum, $B_W$ and $B_X$ denote the network parameters of GNN and the upper-bound constants with respect to the data features $\mathbf{X}$, respectively. $D_{min}$ denotes the minimum degree.*

When the local causal subgraphs are explored, the edge weights in the adjacency matrices are reduced from 1 to $[0, 1]$, the Frobenious norms of $||\mathbf{A}_k||_F$ and $||\mathbf{A}_l||_F$ are decreased, then the bound of generalization error with respect to the global hypothesis can be shrunk. Hence, the generalization ability of global model learned by the proposed FedATH is enhanced. The detailed proof process refers to the Appendix.

## 6. Experiments

### 6.1. Datasets

The comparative experiments are conducted on six real-world graph datasets covering four categories. **Cora**, **PubMed**, and **ogbn-arxiv** (Yang et al., 2016) are three kinds of citation networks, depicting the citation relationships between various papers. **Photo** (Shchur et al., 2018) is a co-purchase network, recording items that are purchased together. **WikiCS** (Mernyei & Cangea, 2020) is a Wiki-page network and constructed based on Wikipedia, recording the relationships between diverse computer science subjects based on hyperlink. **Roman-empire** (Platonov et al., 2023) is an article syntax network via counting the Roman Empire. The details of above six datasets are reported in Table 7. To simulate the distributed graphs, the Louvain method (Blondel et al., 2008) is used to separate the complete graph to multiple clients, e.g., 10, 15, 20.

### 6.2. Compared Methods

We compare the proposed FedATH with nine federated learning algorithms, including conventional and graph-oriented methods. **FedAvg** (McMahan et al., 2017) is used as the baseline. **FedProx** (Li et al., 2020), **MOON** (Li et al., 2021), **FedOPT** (Reddi et al., 2021), and **FedProto** (Tan et al., 2022) are four conventional federated learning approaches, effectively coping with the common distributed data. **FedSage+** (Zhang et al., 2021), **FGSSL** (Huang et al., 2023a), **FedPUB** (Baek et al., 2023), **FedTAD** (Zhu et al.,

| Type | Method | Cora | | | PubMed | | | ogbn-arxiv | | |
|------|--------|------|------|------|------|------|------|------|------|------|
| | | K = 10 | K = 15 | K = 20 | K = 10 | K = 15 | K = 20 | K = 10 | K = 15 | K = 20 |
| BL | FedAvg | 73.59 | 69.52 | 62.44 | 82.42 | 81.51 | 80.80 | 35.75 | 34.74 | 33.57 |
| FL | FedProx | 74.29 | 69.96 | 63.19 | 82.43 | 81.55 | 80.82 | 35.67 | 34.35 | 33.62 |
| | MOON | 74.22 | 70.46 | 61.35 | 82.49 | 81.57 | 80.25 | 34.89 | 33.69 | 33.88 |
| | FedOPT | 74.56 | 71.31 | 63.71 | 81.52 | 80.45 | 80.92 | 36.42 | 34.41 | 35.53 |
| | FedProto | 74.56 | 70.04 | 63.36 | 82.70 | 81.54 | 80.82 | 35.84 | 34.56 | 33.57 |
| FGL | FedSage+ | 73.98 | 67.35 | 65.00 | 82.36 | 78.23 | 78.66 | 41.02 | 36.64 | 37.21 |
| | FGSSL | 74.47 | 72.21 | 65.89 | 82.38 | 81.86 | 80.99 | 39.22 | 36.61 | 35.20 |
| | FedPUB | 75.35 | 72.43 | 66.44 | 82.67 | 79.06 | 79.60 | 39.02 | 36.87 | 36.41 |
| | FedTAD | 74.29 | 72.23 | 63.74 | 82.72 | 82.03 | 81.19 | 37.65 | 36.28 | 34.65 |
| FGL | FedATH | **77.90** | **73.42** | **67.97** | **84.06** | **83.61** | **83.03** | **42.21** | **38.46** | **39.54** |

*Table 1.* Performance comparison (ACC %) on Cora, PubMed, and ogbn-arxiv datasets for all compared methods, where BL denotes the baseline, the optimal results are **bolded** and the suboptimal results are underlined.

| Type | Method | Photo | | | WikiCS | | | Roman-empire | | |
|------|--------|------|------|------|------|------|------|------|------|------|
| | | K = 10 | K = 15 | K = 20 | K = 10 | K = 15 | K = 20 | K = 10 | K = 15 | K = 20 |
| BL | FedAvg | 87.19 | 86.04 | 84.35 | 69.56 | 66.33 | 67.54 | 34.41 | 33.83 | 32.23 |
| FL | FedProx | 87.12 | 86.49 | 84.35 | 69.47 | 66.22 | 67.57 | 34.30 | 33.64 | 32.04 |
| | MOON | 86.45 | 85.24 | 81.47 | 70.03 | 67.16 | 66.53 | 33.97 | 33.78 | 33.03 |
| | FedOPT | 87.73 | 86.10 | 84.20 | 69.39 | 68.11 | 66.46 | 34.35 | 33.44 | 32.10 |
| | FedProto | 87.32 | 87.86 | 84.82 | 70.04 | 66.97 | 68.16 | 34.93 | 33.64 | 32.26 |
| FGL | FedSage+ | 88.46 | 86.80 | 84.37 | 71.72 | 69.32 | 70.69 | 41.59 | 39.35 | 39.12 |
| | FGSSL | 88.56 | 86.17 | 84.27 | 70.63 | 68.97 | 68.64 | 36.96 | 36.85 | 35.21 |
| | FedPUB | 88.79 | 87.17 | 84.21 | 72.30 | 69.82 | 69.88 | 37.31 | 36.12 | 35.47 |
| | FedTAD | 87.76 | 86.60 | 84.94 | 71.63 | 69.32 | 69.01 | 39.01 | 37.94 | 37.28 |
| FGL | FedATH | **90.33** | **88.61** | **85.50** | **75.22** | **72.16** | **71.38** | **48.18** | **47.19** | **45.26** |

*Table 2.* Performance comparison (ACC %) on Photo, WikiCS, and Roman-empire datasets for all compared methods, where BL denotes the baseline, the optimal results are **bolded** and the suboptimal results are underlined.

2024) are four graph-oriented federated learning approaches, tailored for the distributed graph data.

### 6.3. Implementation Details

For the backbone of causal and biased GNNs, a 2-layer graph convolutional network is adopted, encoding the raw graph data into 64-dimensional embedding. The Adam is employed as optimizer with the learning rate set as 0.001. The numbers of communication rounds and local training epochs are fixed as 100 and 3, respectively. Considering the node classification task, classification accuracy (ACC) and F1-score (F1) are employed as the evaluation metrics. For the trade-off parameter $\lambda$, it is tuned in range of $\{0.001, 0.1, 7, 10\}$.

### 6.4. Performance Comparison

The experimental results for all compared methods are reported in Tables 1 and 2, where three cases of client number are considered, i.e., 10, 15, 20. Undoubtedly, FedAvg achieves inferior performance, indicating that simply aggre-

gating model parameters is not effective against the topological heterogeneity. It can also be seen that the conventional federated learning algorithms achieve minor gains versus FedAvg, and are even less effective than FedAvg in many cases, showing that they cannot cope well with the topology heterogeneity in FGL. Conversely, the tailored FGL methods achieve relatively good results, which thanks to their strategies for heterogeneity of graph data such as fixing the global graph or generating the global representations. However, the proposed FedATH achieves the optimal results over the other compared approaches, demonstrating that using less key information is more conducive to more gains for the performance.

### 6.5. Ablation Study

**Effects of Different Losses.** In addition to the cross entropy loss, two important losses are included in the proposed FedATH: $\mathcal{L}_{ENT}$ and $\mathcal{L}_{DEP}$. The tailored ablation experiments are designed to verify their effects in Tables 3 and 4. It can be seen that when either item is removed, the

performance of FedATH is inevitably weakened, suggesting that both of them play an essential role in achieving the separation of causal and biased subgraphs. Only when both losses are present, the proposed FedATH reaches the optimal results.

**Effects of Sharing Different Components.** Overall, three network modules are equipped in each client, including the edge evaluator, the causal GNN, and the biased GNN. We test the performance with sharing various components. As shown in Tables 5 and 6, the proposed FedATH reaches its optimality when only the local CGs are shared in general. When local EEs and BGs are involved in sharing, the performance suffers from varying degrees of degradation. This is because they capture the biased information caused by local topology heterogeneity, which is not conducive to model generalization.

| $\mathcal{L}_{ENT}$ | $\mathcal{L}_{DEP}$ | Cora | | | PubMed | | |
|---|---|---|---|---|---|---|---|
| | | K = 10 | K = 15 | K = 20 | K = 10 | K = 15 | K = 20 |
| ✗ | ✗ | 73.59 | 69.52 | 62.44 | 82.42 | 81.51 | 80.80 |
| ✗ | ✓ | 76.40 | 72.92 | 66.29 | 83.78 | 83.29 | 82.44 |
| ✓ | ✗ | 75.09 | 72.12 | 63.01 | 83.95 | 83.36 | 82.42 |
| ✓ | ✓ | **77.90** | **73.42** | **67.97** | **84.06** | **83.61** | **83.03** |

*Table 3.* Ablation results (ACC %) with respect to two principal losses on Cora and PubMed datasets.

| $\mathcal{L}_{ENT}$ | $\mathcal{L}_{DEP}$ | Photo | | | WikiCS | | |
|---|---|---|---|---|---|---|---|
| | | K = 10 | K = 15 | K = 20 | K = 10 | K = 15 | K = 20 |
| ✗ | ✗ | 87.19 | 86.04 | 84.35 | 69.56 | 66.33 | 68.73 |
| ✗ | ✓ | 89.24 | 85.01 | 85.06 | 72.07 | 70.00 | 70.47 |
| ✓ | ✗ | 89.81 | 86.66 | 85.24 | 72.92 | 69.86 | 71.15 |
| ✓ | ✓ | **90.33** | **88.61** | **85.50** | **75.22** | **72.16** | **71.38** |

*Table 4.* Ablation results (ACC %) with respect to two principal losses on Photo and WikiCS datasets.

| Sharing | Cora | | | PubMed | | |
|---|---|---|---|---|---|---|
| | K = 10 | K = 15 | K = 20 | K = 10 | K = 15 | K = 20 |
| $CG+EE$ | **78.69** | 73.34 | 67.02 | 83.23 | 83.01 | 81.77 |
| $CG+BG$ | 76.59 | 73.17 | 66.45 | 83.16 | 82.94 | 82.53 |
| $CG+EV+BG$ | 78.25 | **73.94** | 67.19 | 83.18 | 83.13 | 81.74 |
| $CG$ | 77.90 | 73.42 | **67.97** | **84.06** | **83.61** | **83.03** |

*Table 5.* The performance comparison (ACC %) with various shared components on Cora and PubMed datasets, where $CG$, $BG$, and $EE$ denote the causal GNN, biased GNN, and edge evaluator, respectively.

### 6.6. Hyperparameter Study

The trade-off parameter $\lambda$ balances the contribution of HSIC loss, its importance is validated by tuning the values in $\{0.0001, 0.001, ..., 10\}$. From Fig. 4, we can observe that a larger $\lambda$ is required on Cora dataset while a smaller one

| Sharing | Photo | | | WikiCS | | |
|---|---|---|---|---|---|---|
| | K = 10 | K = 15 | K = 20 | K = 10 | K = 15 | K = 20 |
| $CG+EE$ | 86.96 | 85.20 | 84.78 | 74.55 | 71.27 | 67.45 |
| $CG+BG$ | 86.87 | 85.58 | 84.86 | 74.21 | 70.86 | 69.48 |
| $CG+EE+BG$ | 86.61 | 85.62 | 84.90 | 74.77 | 71.57 | 67.40 |
| $CG$ | **90.33** | **88.61** | **85.50** | **75.22** | **72.16** | **71.38** |

*Table 6.* The performance comparison (ACC %) with various shared components on Photo and WikiCS datasets, where $CG$, $BG$, and $EE$ denote the causal GNN, biased GNN, and edge evaluator, respectively.

is set on WikiCS dataset. Different datasets have different statistical characteristics, and an appropriate $\lambda$ is needed to achieve a well separation of causal subgraph and biased subgraph. Hence, it is necessary to introduce $\lambda$. Furthermore, the impact of label ratio is verified in Fig. 5. The larger the label ratio, the higher the performance is obtained for all algorithms. Fortunately, the proposed FedATH still maintains leading performance.

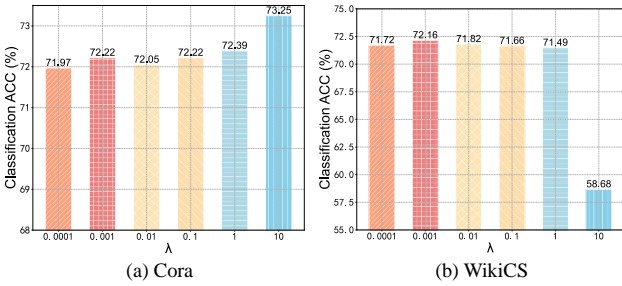

(a) Cora  (b) WikiCS

*Figure 4.* The performance comparison when the trade-off parameter $\lambda$ is tuned, where the client number is set as 15.

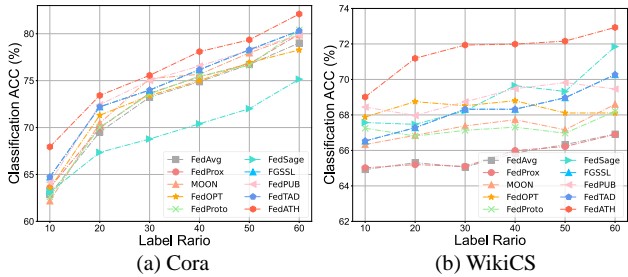

(a) Cora  (b) WikiCS

*Figure 5.* The performance comparison for all compared methods with different label ratios, where the client number is set as 15.

## 7. Conclusion

In this paper, we propose a new FGL method called FedATH to cope with the topology heterogeneity across different local graphs. We recognize that correcting model training by creating increments could be tough. Instead, we mitigate the topology heterogeneity via reducing the superfluous information. Specifically, the local edge evaluators are developed

to distinguish the local graphs into the causal subgraphs and biased subgraphs. A dual-GNN architecture maps the two subgraphs into latent representations. With the aid of the designed losses, the separability of the two subgraphs is enhanced. The experimental results verify the advancement of FedATH over other compared methods. However, in real-world scenarios, node labels are often unavailable, and the lack of unified semantic information further exacerbates the difficulty of federation training. In future work, we will explore how to achieve effective federated graph learning in unsupervised scenarios.

## Acknowledgement

The research is supported by the National Key R&D Program of China (2023YFB2703700), the National Natural Science Foundation of China (62176269).

## Impact Statement

This paper presents work whose goal is to advance the field of Machine Learning. There are many potential societal consequences of our work, none which we feel must be specifically highlighted here.

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

## A. Summary of Appendix

In the appendix, the following contents are included.

- Summaries of the proposed FedATH in Algorithm 1 and the used datasets.

- Performance comparison with F1 metric.

- Proof of Theorem 5.1 and Corollary 5.2.

- Performance with increasing client number.

- Convergence verification of the proposed FedATH.

## B. Summaries of The Propose FedATH and The Used Datasets

We summarize the main steps of the proposed FedATH in Algorithm 1 and the main statistics of used graph datasets in Table 7.

---

**Algorithm 1** The flow of FedATH

---

**Input:** Number of clients $K$, local training epochs $E$, communication rounds $T$, learning rate $\eta$, trade-off parameter $\lambda$, local graph data $G_k = (V_k, E_k, \mathbf{X}_k)$, causal GNN $f_k^c$, and biased GNN $f_k^b$.
**Output:** Global causal GNN $f^c$.
 1: **Client Side:**
 2: **for** $k = 1 : K$ **in parallel do**
 3:     **for** epoch $e = 1 : E$ **do**
 4:         Calculate the casual and biased edge mask matrices $\mathbf{\Omega}_k^c$ and $\mathbf{\Omega}_k^b$ via Eqs. (5) and (6);
 5:         Calculate the casual and biased representations $\mathbf{H}_k^c$ and $\mathbf{H}_k^b$ via Eq. (7);
 6:         Calculate $\mathcal{L}_{CE} \leftarrow (\mathbb{1}_{\mathbf{y}_i}, \mathbf{h}_i^c)$ via Eq. (8);
 7:         Calculate $\mathcal{L}_{ENT} \leftarrow (\mathbf{h}_i^b)$ via Eq. (9);
 8:         Calculate $\mathcal{L}_{DEP} \leftarrow (\mathbf{H}^c, \mathbf{H}^b)$ via Eq. (11);
 9:         Update $f_k^{c,e} \leftarrow f_k^{c,e-1} - \eta\nabla(\mathcal{L}_{CE} + \mathcal{L}_{DEP})$;
10:         Update $f_k^{b,e} \leftarrow f_k^{b,e-1} - \eta\nabla(\mathcal{L}_{ENT} + \mathcal{L}_{DEP})$;
11:     **end for**
12:     Upload the local causal GNN $f_k^c$ to the server;
13: **end for**
14: **Server Side:**
15: **for** $t = 1 : T$ **do**
16:     Aggregate the parameter of global causal GNN via $\mathbf{W}^{c,t} \leftarrow \sum_{k=1}^K N_k/N\mathbf{W}_k^{c,t-1}$;
17:     Distribute the global causal GNN $\mathbf{W}^{c,t}$ to clients;
18: **end for**

---

| Dataset | Nodes | Features | Edges | Classes | Train / Val / Test |
|---|---|---|---|---|---|
| Cora | 2,708 | 1,433 | 5,429 | 7 | 20% / 40% / 40% |
| PubMed | 19,717 | 500 | 44,338 | 3 | 20% /40 % / 40% |
| ogbn-arxiv | 169,343 | 128 | 231,559 | 40 | 60% / 20% / 20% |
| Photo | 7,487 | 745 | 119,043 | 8 | 20% / 40% / 40% |
| WikiCS | 11,701 | 300 | 216,123 | 10 | 50% / 20% / 30% |
| Roman-empire | 22,662 | 300 | 32,927 | 18 | 50% / 20% / 30% |

*Table 7.* Descriptions of six graph datasets.

| Type | Method | Cora | | | PubMed | | | ogbn-arxiv | | |
|------|--------|------|------|------|--------|------|------|------------|------|------|
| | | K = 10 | K = 15 | K = 20 | K = 10 | K = 15 | K = 20 | K = 10 | K = 15 | K = 20 |
| BL | FedAvg | 72.04 | 66.22 | 58.80 | 80.20 | 79.00 | 78.87 | 26.05 | 26.17 | 25.07 |
| FL | FedProx | 72.89 | 66.68 | 59.65 | 80.21 | 79.05 | 78.88 | 26.07 | 25.93 | 25.12 |
| | MOON | 71.88 | 67.95 | 58.03 | 80.41 | 79.10 | 78.10 | 25.82 | 24.73 | 25.56 |
| | FedOPT | 67.73 | 63.64 | 59.75 | 78.10 | 75.33 | 76.51 | 24.06 | 24.23 | 26.35 |
| | FedProto | 72.99 | 67.01 | 59.38 | 80.44 | 79.03 | 78.88 | 26.09 | 25.99 | 24.99 |
| FGL | FedSage+ | 71.96 | 62.07 | 59.85 | 82.18 | 78.03 | 76.18 | 27.45 | 28.82 | 30.23 |
| | FGSSL | 72.94 | 69.33 | 61.85 | 79.88 | 79.32 | 78.82 | 30.41 | 27.93 | 26.59 |
| | FedPUB | 73.08 | 66.76 | 60.01 | 80.43 | 75.44 | 79.63 | 29.55 | 24.83 | 24.37 |
| | FedTAD | 71.57 | 70.38 | 60.34 | 80.26 | 78.84 | 78.57 | 27.20 | 26.74 | 25.10 |
| FGL | FedATH | **76.93** | **72.16** | **65.80** | **82.71** | **81.82** | **81.31** | **34.90** | **30.94** | **31.97** |

*Table 8.* Performance comparison (F1 %) on Cora, PubMed, and ogbn-arxiv datasets for all compared methods, where BL denotes the baseline, the optimal results are **bolded** and the suboptimal results are underlined.

| Type | Method | Photo | | | WikiCS | | | Roman-empire | | |
|------|--------|-------|------|------|--------|------|------|--------------|------|------|
| | | K = 10 | K = 15 | K = 20 | K = 10 | K = 15 | K = 20 | K = 10 | K = 15 | K = 20 |
| BL | FedAvg | 84.85 | 83.06 | 81.37 | 63.65 | 61.66 | 62.38 | 30.34 | 29.74 | 27.80 |
| FL | FedProx | 84.78 | 83.70 | 81.37 | 63.55 | 61.50 | 62.34 | 30.22 | 29.52 | 27.60 |
| | MOON | 83.64 | 82.02 | 79.43 | 63.99 | 62.15 | 60.90 | 29.86 | 29.60 | 28.55 |
| | FedOPT | 84.92 | 83.02 | 80.26 | 60.53 | 60.37 | 56.86 | 30.50 | 29.58 | 28.08 |
| | FedProto | 84.94 | 85.07 | 81.70 | 64.04 | 61.95 | 62.25 | 30.01 | 28.82 | 27.32 |
| FGL | FedSage+ | 86.13 | 82.00 | 82.94 | 61.47 | 62.17 | 61.15 | 31.88 | 29.39 | 29.60 |
| | FGSSL | 85.99 | 82.62 | 80.24 | 64.84 | 64.11 | 62.91 | 33.59 | 33.08 | 31.52 |
| | FedPUB | 85.86 | 83.53 | 79.32 | 63.79 | 59.18 | 59.71 | 30.15 | 29.22 | 28.71 |
| | FedTAD | 85.24 | 83.45 | 81.20 | 65.31 | 63.69 | 63.10 | 35.08 | 33.80 | 33.21 |
| FGL | FedATH | **88.80** | **86.52** | **82.88** | **71.10** | **68.15** | **67.37** | **44.98** | **44.27** | **42.08** |

*Table 9.* Performance comparison (F1 %) on Photo, WikiCS, and Roman-empire datasets for all compared methods, where BL denotes the baseline, the optimal results are **bolded** and the suboptimal results are underlined.

## C. Performance Comparison with F1

To avoid the biased evaluation for experimental results, we employ an additional widely used classification metric F1-score (F1), which can overcome the evaluation distortion caused by the unbalanced data. From Tables 8 and 9, it can be seen that the proposed FedTAH consistently achieves the optimal results just like ACC, showing that FedATH solidly enhances the ability of global model. Nevertheless, some algorithms achieve suboptimality on ACC but not on F1, such as FGSSL and FedTAD, demonstrating that they are vulnerable to unbalanced data.

## D. Proof of Theorem 1 and Corollary 1

For **Theorem** 5.1, we provide the detailed proof process. Denote $\mathcal{D}$ and $\mathcal{D}_k$ the global and local distributions, respectively. $\tilde{\mathcal{D}}_k$ denotes the empirical local distribution. $h_k$ is the local hypothesis learned on the local empirical distribution $\tilde{\mathcal{D}}_k$ and defined as $h_k : \mathcal{X} \to \mathcal{Y}$, mapping the data features into predicted labels. $h = 1/K \sum_{k=1}^{K} h_k$ denotes the global hypothesis integrated by local hypothesis. $\mathcal{H}$ denotes the hypothesis space of VC-dimension $d$. Moreover, without losing generality, it is specified that $N_1 = ... = N_K = m$.

**Theorem D.1.** *Given an FGL system with global distribution $\mathcal{D}$ and local distribution $\mathcal{D}_k$, with the the probability at least*

$1 - \delta$ $(0 < \delta \leq 1)$, the generalization error for any hypothesis $h_k$ satisfies

$$
\begin{aligned}
\mathcal{R}_{\mathcal{D}}(h) \leq & \frac{1}{K} \sum_{k \in [K]} \hat{\mathcal{R}}_{\tilde{\mathcal{D}}_k}(h_k) \\
& + \frac{1}{K(K-1)} \sum_{k \in [K]} \sum_{l \neq k}^{K} d_{\mathcal{H} \Delta \mathcal{H}}\left(\tilde{\mathcal{D}}_k, \tilde{\mathcal{D}}_l\right) + \epsilon \\
& + \frac{1}{K} \sum_{k \in [K]} \lambda_k + \sqrt{\frac{4}{m}\left(d \log \frac{2em}{d} + \log \frac{4K}{\delta}\right)},
\end{aligned}
\tag{15}
$$

where $\hat{\mathcal{R}}_{\tilde{\mathcal{D}}_k}(h_k)$ denotes the empirical risk on $\tilde{\mathcal{D}}_k$, $d_{\mathcal{H} \Delta \mathcal{H}}(\tilde{\mathcal{D}}_k, \tilde{\mathcal{D}}_l)$ is the $\mathcal{H}$-distance between $\tilde{\mathcal{D}}_k$ and $\tilde{\mathcal{D}}_l$, $\epsilon$ denotes a upper-bound constant with respect to $d_{\mathcal{H} \Delta \mathcal{H}}(\tilde{\mathcal{D}}_l, \tilde{\mathcal{D}}), \forall l \in [K]$, $\lambda_k = \min_h(\mathcal{R}_{\mathcal{D}}(h) + \mathcal{R}_{\mathcal{D}_k}(h))$ denotes the optimal risk on $\mathcal{D}$ and $\mathcal{D}_k$.

**Corollary D.2.** *Given an FGL system, the $k$-th and $l$-th empirical local distributions are denoted as $\tilde{\mathcal{D}}_k$ and $\tilde{\mathcal{D}}_l$, its generalization error follows **Theorem** 5.1. For $d_{\mathcal{H} \Delta \mathcal{H}}(\tilde{\mathcal{D}}_k, \tilde{\mathcal{D}}_l)$, the following inequality holds.*

$$
\begin{aligned}
d_{\mathcal{H} \Delta \mathcal{H}}(\tilde{\mathcal{D}}_k, \tilde{\mathcal{D}}_l) \leq & 1 + \sup_f B_W^2 B_X \left(\frac{1}{D_{\min} m} + \frac{1}{D_{\min}^2}\right) \\
& (\|\mathbf{A}_k\|_F + \|\mathbf{A}_l\|_F) + \frac{B_W^2}{m} \|\mathbf{X}_k - \mathbf{X}_l\|_F,
\end{aligned}
\tag{16}
$$

where $\sup$ *denotes the supremum, $B_W$ and $B_X$ denote the network parameters of GNN and the upper-bound constants with respect to the data features $\mathbf{X}$, respectively. $D_{min}$ denotes the minimum degree.*

When the local causal subgraphs are explored, the edge weights in the adjacency matrices are reduced from 1 to $[0, 1]$, the Frobenious norms of $\|\mathbf{A}_k\|_F$ and $\|\mathbf{A}_l\|_F$ are decreased, then the bound of generalization error with respect to the global hypothesis can be shrunk. Hence, the generalization ability of global model learned by the proposed FedATH is enhanced.

*Proof.* According to the generalization error of FL (Barnes et al., 2022; Zhu et al., 2021), we first introduce following **Lemma D.3**.

**Lemma D.3.** *Given an FL system with global distribution $\mathcal{D}$ and local distribution $\mathcal{D}_k$, the generalization error for any hypothesis with the probability at least $1 - \delta$ $(0 < \delta \leq 1)$ is*

$$
\mathcal{R}_{\mathcal{D}}(h) \leq \frac{1}{K} \sum_{k \in [K]} \hat{\mathcal{R}}_{\tilde{\mathcal{D}}_k}(h_k) + \frac{1}{K} \sum_{k \in [K]} \left(d_{\mathcal{H} \Delta \mathcal{H}}\left(\tilde{\mathcal{D}}_k, \tilde{\mathcal{D}}\right) + \lambda_k\right) + \sqrt{\frac{4}{m}\left(d \log \frac{2em}{d} + \log \frac{4K}{\delta}\right)}.
\tag{17}
$$

For the $\mathcal{H}$-divergence $d_{\mathcal{H} \Delta \mathcal{H}}(\tilde{\mathcal{D}}_k, \tilde{\mathcal{D}})$, we further have

$$
\begin{aligned}
d_{\mathcal{H} \Delta \mathcal{H}}\left(\tilde{\mathcal{D}}_k, \tilde{\mathcal{D}}\right) \leq & d_{\mathcal{H} \Delta \mathcal{H}}\left(\tilde{\mathcal{D}}_k, \tilde{\mathcal{D}}_l\right) + d_{\mathcal{H} \Delta \mathcal{H}}\left(\tilde{\mathcal{D}}_l, \tilde{\mathcal{D}}\right) \\
(K-1) d_{\mathcal{H} \Delta \mathcal{H}}\left(\tilde{\mathcal{D}}_k, \tilde{\mathcal{D}}\right) \leq & \sum_{l \neq k}^{K} \left[d_{\mathcal{H} \Delta \mathcal{H}}\left(\tilde{\mathcal{D}}_k, \tilde{\mathcal{D}}_l\right) + d_{\mathcal{H} \Delta \mathcal{H}}\left(\tilde{\mathcal{D}}_l, \tilde{\mathcal{D}}\right)\right]
\end{aligned}
\tag{18}
$$

Assume $\forall l \in [K], d_{\mathcal{H} \Delta \mathcal{H}}(\tilde{\mathcal{D}}_l, \tilde{\mathcal{D}}) \leq \epsilon_0$, it can obtain

$$
\begin{aligned}
(K-1) d_{\mathcal{H} \Delta \mathcal{H}}\left(\tilde{\mathcal{D}}_k, \tilde{\mathcal{D}}\right) &\leq \sum_{l \neq k}^{K}\left[d_{\mathcal{H} \Delta \mathcal{H}}\left(\tilde{\mathcal{D}}_k, \tilde{\mathcal{D}}_l\right)+\epsilon_0\right] \\
\frac{1}{K} \sum_{k \in[K]} d_{\mathcal{H} \Delta \mathcal{H}}\left(\tilde{\mathcal{D}}_k, \tilde{\mathcal{D}}\right) &\leq \frac{1}{K(K-1)} \sum_{k \in[K]} \sum_{l \neq k}^{K}\left[d_{\mathcal{H} \Delta \mathcal{H}}\left(\tilde{\mathcal{D}}_k, \tilde{\mathcal{D}}_l\right)+\epsilon_0\right] \\
&\leq \frac{1}{K(K-1)} \sum_{k \in[K]} \sum_{l \neq k}^{K} d_{\mathcal{H} \Delta \mathcal{H}}\left(\tilde{\mathcal{D}}_k, \tilde{\mathcal{D}}_l\right) \\
&+ \underbrace{\frac{1}{K(K-1)} \sum_{k \in[K]} \sum_{l \neq k}^{K} \epsilon_0}_{\epsilon} .
\end{aligned}
\tag{19}
$$

Then, we have

$$
\begin{aligned}
\mathcal{R}_{\mathcal{D}}(h) \leq & \frac{1}{K} \sum_{k \in[K]} \hat{\mathcal{R}}_{\tilde{\mathcal{D}}_k}\left(h_k\right)+\frac{1}{K(K-1)} \sum_{k \in[K]} \sum_{l \neq k}^{K} d_{\mathcal{H} \Delta \mathcal{H}}\left(\tilde{\mathcal{D}}_k, \tilde{\mathcal{D}}_l\right)+\epsilon \\
& +\frac{1}{K} \sum_{k \in[K]} \lambda_k+\sqrt{\frac{4}{m}\left(d \log \frac{2 e m}{d}+\log \frac{4 K}{\delta}\right)} .
\end{aligned}
\tag{20}
$$

The proof completes. $\square$

*Proof.* Next, we estimate the upper bound of $d_{\mathcal{H} \Delta \mathcal{H}}(\tilde{\mathcal{D}}_k, \tilde{\mathcal{D}}_l)$ in the context of FGL. $m$ nodes with their topology from the distributions of the $k$-th and $l$-th clients are sampled, respectively: $G_k=(\mathbf{A}_k, \mathbf{X}_k) \sim \tilde{\mathcal{D}}_k$, $G_l=(\mathbf{A}_l, \mathbf{X}_l) \sim \tilde{\mathcal{D}}_l$. Taking a two-layer GCN as an example, it has $f(\mathbf{A}, \mathbf{X})=\operatorname{sigmoid}(\mathbf{P} \sigma(\mathbf{P X W}_1) \mathbf{W}_2)$, where $\mathbf{P}=(\mathbf{D}+\mathbf{I})^{-1/2}(\mathbf{A}+\mathbf{I})(\mathbf{D}+\mathbf{I})^{-1/2}$, $\mathbf{W}_1$ and $\mathbf{W}_2$ denote the parameters of first and second layers. Further, Suppose $\|\mathbf{X}\| \leq B_X$, $\|\mathbf{W}_1\| \leq B_W$, and $\|\mathbf{W}_2\| \leq B_W$, then we have

$$
\begin{aligned}
& \left\|f\left(G_k\right)-f\left(G_l\right)\right\|_F \\
= & \left\|\operatorname{sigmoid}\left(\mathbf{P}_k \sigma\left(\mathbf{P}_k \mathbf{X}_k \mathbf{W}_1\right) \mathbf{W}_2\right)-\operatorname{sigmoid}\left(\mathbf{P}_l \sigma\left(\mathbf{P}_l \mathbf{X}_l \mathbf{W}_1\right) \mathbf{W}_2\right)\right\|_F \\
\leq & \left\|\mathbf{P}_k \sigma\left(\mathbf{P}_k \mathbf{X}_k \mathbf{W}_1\right) \mathbf{W}_2-\mathbf{P}_l \sigma\left(\mathbf{P}_l \mathbf{X}_l \mathbf{W}_1\right) \mathbf{W}_2\right\|_F \\
\leq & \left\|\mathbf{W}_2\right\|_F\left\|\mathbf{W}_1\right\|_F\left(\left\|\mathbf{X}_k-\mathbf{X}_l\right\|_F+\left\|\mathbf{P}_k-\mathbf{P}_l\right\|_F\left\|\mathbf{X}_1\right\|_F\right) \\
\leq & B_W^2\left\|\mathbf{X}_k-\mathbf{X}_l\right\|_F+B_W^2 B_X\left\|\mathbf{P}_k-\mathbf{P}_l\right\|_F \\
\leq & B_W^2 B_X\left\|\mathbf{P}_k-\mathbf{P}_l\right\|_F+B_W^2\left\|\mathbf{X}_k-\mathbf{X}_l\right\|_F .
\end{aligned}
\tag{21}
$$

For $d_{\mathcal{H} \Delta \mathcal{H}}(\tilde{\mathcal{D}}_k, \tilde{\mathcal{D}}_l)$, its supremum can be written as

$$
\begin{aligned}
& d_{\mathcal{H} \Delta \mathcal{H}}\left(\tilde{\mathcal{D}}_k, \tilde{\mathcal{D}}_l\right) \\
= & 2 \sup_{h_k \in \mathcal{H}, h_l \in \mathcal{H}}\left|\operatorname{Pr}_{\tilde{\mathcal{D}}_k}\left[z_k: h_k(z_k) \neq h_l\left(z_k\right)\right]-\operatorname{Pr}_{\tilde{\mathcal{D}}_l}\left[z_l: h_k\left(z_l\right) \neq h_l\left(z_l\right)\right]\right| \\
= & 2 \sup_{h \in \mathcal{H} \Delta \mathcal{H}}\left|\operatorname{Pr}_{\tilde{\mathcal{D}}_k}\left[z: h\left(z_k\right)=1\right]-\operatorname{Pr}_{\tilde{\mathcal{D}}_l}\left[z: h\left(z_l\right)=1\right]\right| \\
= & 2 \sup_{h \in \mathcal{H} \Delta \mathcal{H}}\left|\mathbb{E}_{\tilde{\mathcal{D}}_k} h\left(z_k\right)-\mathbb{E}_{\tilde{\mathcal{D}}_l} h\left(z_l\right)\right|
\end{aligned}
\tag{22}
$$

According to the inequality $0.5\mathbb{E}[h(z)] \leq \mathbb{E}[f(z)] \leq 0.5 + 0.5\mathbb{E}[h(z)]$, it can be further derived

$$
\begin{aligned}
&d_{\mathcal{H}\Delta\mathcal{H}}\left(\tilde{\mathcal{D}}_k, \tilde{\mathcal{D}}_l\right) \\
&\leq 2\sup_f \left|0.5 + 0.5\mathbb{E}_{\tilde{\mathcal{D}}_k} f(z_k) - 0.5\mathbb{E}_{\tilde{\mathcal{D}}_l} f(z_l)\right| \\
&\leq 1 + \sup_f \left|\mathbb{E}_{\tilde{\mathcal{D}}_k} f(z_k) - \mathbb{E}_{\tilde{\mathcal{D}}_l} f(z_l)\right|.
\end{aligned}
\tag{23}
$$

Further, the empirical $\mathcal{H}$-divergence is upper bounded as

$$
\begin{aligned}
&\hat{d}_{\mathcal{H}\Delta\mathcal{H}}\left(\tilde{\mathcal{D}}_k, \tilde{\mathcal{D}}_l\right) \\
&= 1 + \sup_f \left|\frac{1}{m}\sum_i^m f(z_{k,i}) - \frac{1}{m}\sum_j^m f(z_{l,j})\right| \\
&\leq 1 + \frac{1}{m}\sup_f \left|\sum_i^m f(z_{k,i}) - f(z_{l,i})\right| \\
&\leq 1 + \frac{1}{m}\sup_f \sum_i^m \left|f(z_{k,i}) - f(z_{l,i})\right| \\
&= 1 + \frac{1}{m}\sup_f \left\|f(G_k) - f(G_l)\right\|_{1,ele} \\
&\leq 1 + \frac{B_{rank}}{m}\sup_f \left\|f(G_k) - f(G_l)\right\|_F
\end{aligned}
\tag{24}
$$

Notably, the entrywise 1-norm $||\mathbf{A}||_{1,ele}$ adheres to

$$
||\mathbf{A}||_F \leq ||\mathbf{A}||_{1,ele} \leq \sqrt{rank(\mathbf{A})}||\mathbf{A}||_F.
\tag{25}
$$

$B_{rank} = \sqrt{rank(f(G_k) - f(G_l))}$ is a constant.

From Eq. (24), we can see that the upper bound of $\hat{d}_{\mathcal{H}\Delta\mathcal{H}}(\tilde{\mathcal{D}}_k, \tilde{\mathcal{D}}_l)$ depends on $\sup_f ||f(G_k) - f(G_l)||_F$. Recall Eq. (21), $\sup_f ||f(G_k) - f(G_l)||_F$ is bounded by $||\mathbf{P}_k - \mathbf{P}_l||_F$. We further have

$$
\begin{aligned}
&\mathbf{P}_k - \mathbf{P}_l \\
&= \mathbf{D}_k^{-1/2}\mathbf{A}_k\mathbf{D}_k^{-1/2} - \mathbf{D}_l^{-1/2}\mathbf{A}_l\mathbf{D}_l^{-1/2} \\
&= \mathbf{D}_k^{-1/2}\mathbf{A}_k\mathbf{D}_k^{-1/2} - \mathbf{D}_k^{-1/2}\mathbf{A}_l\mathbf{D}_k^{-1/2} \\
&\quad + \mathbf{D}_k^{-1/2}\mathbf{A}_l\mathbf{D}_k^{-1/2} - \mathbf{D}_l^{-1/2}\mathbf{A}_l\mathbf{D}_l^{-1/2} \\
&= \underbrace{\mathbf{D}_k^{-1/2}(\mathbf{A}_k - \mathbf{A}_l)\mathbf{D}_k^{-1/2}}_{\mathbf{T}_1} + \underbrace{\mathbf{D}_k^{-1/2}\mathbf{A}_l\mathbf{D}_k^{-1/2} - \mathbf{D}_l^{-1/2}\mathbf{A}_l\mathbf{D}_l^{-1/2}}_{\mathbf{T}_2}
\end{aligned}
\tag{26}
$$

Then, it can obtain

$$
||\mathbf{P}_k - \mathbf{P}_l||_F \leq ||\mathbf{T}_1||_F + ||\mathbf{T}_2||_F.
\tag{27}
$$

For $||\mathbf{T}_1||_F$ and $||\mathbf{T}_2||_F$, we estimate their upper bound. Denote $D_{min}$ the minimum degree of $\mathbf{D}_k$ and $\mathbf{D}_l$, then it has

$$
\begin{aligned}
&||\mathbf{T}_1||_F = \left\|\mathbf{D}_k^{-1/2}(\mathbf{A}_k - \mathbf{A}_l)\mathbf{D}_k^{-1/2}\right\|_F \\
&||\mathbf{T}_1||_F^2 = \sum_{i,j}\left[\mathbf{D}_{k_{ii}}^{-1/2}(\mathbf{A}_k - \mathbf{A}_l)_{ij}\mathbf{D}_{k_{jj}}^{-1/2}\right]^2 \\
&\leq \left(D_{min}^{-1/2}\right)^4 \sum_{i,j}(\mathbf{A}_k - \mathbf{A}_l)_{ij}^2 = \left(D_{min}^{-1/2}\right)^4 ||\mathbf{A}_k - \mathbf{A}_l||_F^2 \\
&= \frac{1}{D_{min}^2}||\mathbf{A}_k - \mathbf{A}_l||_F^2
\end{aligned}
\tag{28}
$$

$$\mathbf{T}_2 = \mathbf{D}_k^{-1/2}\mathbf{A}_l\mathbf{D}_k^{-1/2} - \mathbf{D}_l^{-1/2}\mathbf{A}_l\mathbf{D}_l^{-1/2}$$
$$= \left[\mathbf{D}_k^{-1/2}-\mathbf{D}_l^{-1/2}\right]\mathbf{A}_l\mathbf{D}_k^{-1/2}+\mathbf{D}_l^{-1/2}\mathbf{A}_l\left[\mathbf{D}_k^{-1/2}-\mathbf{D}_l^{-1/2}\right] \tag{29}$$

Denote $\delta_i$ the $(i,i)$-th element of $\mathbf{D}_k^{-1/2} - \mathbf{D}_l^{-1/2}$. With the Taylor extension, we have

$$\delta_i = \mathbf{D}_{k_{ii}}^{-1/2} - \mathbf{D}_{l_{ii}}^{-1/2} \approx -\frac{1}{2}\mathbf{D}_{k_{ii}}^{-3/2}\left(\mathbf{D}_{k_{ii}} - \mathbf{D}_{l_{ii}}\right)$$

$$\mathbf{D}_{k_{ii}} - \mathbf{D}_{l_{ii}} = -\frac{1}{2}\mathbf{D}_{k_{ii}}^{-3/2}\sum_j \mathbf{A}_{kij} - \mathbf{A}_{lij}. \tag{30}$$

Thus, it can be derived

$$|\delta_i| \leq \frac{1}{2D_{\min}^{3/2}}\left|\sum_j \left(\mathbf{A}_{k_{ij}} - \mathbf{A}_{l_{ij}}\right)\right| \tag{31}$$

For the $(i,j)$-th element of $\mathbf{T}_2$, we can obtain

$$\left|\mathbf{T}_{2_{ij}}\right| \leq |\delta_i| \cdot \left|\mathbf{A}_{l_{ij}}\right| \cdot \mathbf{D}_{k_{jj}}^{-1/2} + \mathbf{D}_{l_{ii}}^{-1/2} \cdot \left|\mathbf{A}_{l_{ij}}\right| \cdot |\delta_j| \tag{32}$$

Since $\mathbf{A}_{l_{ij}} \leq 1$ and the entries of $\mathbf{D}_k^{-1/2}$ and $\mathbf{D}_l^{-1/2}$ are bounded by $D_{min}^{-1/2}$, the following inequality holds:

$$\left|\mathbf{T}_{2_{ij}}\right| \leq |\delta_i| \cdot \left|\mathbf{A}_{l_{ij}}\right| \cdot \mathbf{D}_{k_{jj}}^{-1/2} + \mathbf{D}_{l_{ii}}^{-1/2} \cdot \left|\mathbf{A}_{l_{ij}}\right| \cdot |\delta_j|$$

$$= \frac{1}{2\mathbf{D}_{\min}^{3/2}}\left|\sum_k(\mathbf{A}_{k_{ik}}-\mathbf{A}_{l_{ik}})\right| \cdot 1 \cdot D_{\min}^{-1/2} + D_{\min}^{-1/2} \cdot 1 \cdot \frac{1}{2\mathbf{D}_{\min}^{3/2}}\left|\sum_k\left(\mathbf{A}_{k_{jk}}-\mathbf{A}_{l_{jk}}\right)\right| \tag{33}$$

$$= \frac{1}{2D_{\min}^2}\left(\left|\sum_k\left(\mathbf{A}_{k_{ik}} - \mathbf{A}_{l_{ik}}\right)\right| + \left|\sum_k\left(\mathbf{A}_{k_{jk}} - \mathbf{A}_{l_{jk}}\right)\right|\right)$$

Then, it follows that

$$\|\mathbf{T}_2\|_F^2 = \sum_{i,j}\left|\mathbf{T}_{2_{ij}}\right|^2$$

$$\leq \left(\frac{1}{2D_{\min}^2}\right)^2\sum_{i,j}\left(\left|\sum_k\left(\mathbf{A}_{k_{ik}}-\mathbf{A}_{l_{ik}}\right)\right| + \left|\sum_k\left(\mathbf{A}_{k_{jk}}-\mathbf{A}_{l_{jk}}\right)\right|\right)^2$$

$$\leq \left(\frac{1}{2D_{\min}^2}\right)^2 2\sum_{i,j}\left(\left|\sum_k\left(\mathbf{A}_{k_{ik}}-\mathbf{A}_{l_{ik}}\right)\right|^2 + \left|\sum_k\left(\mathbf{A}_{k_{jk}}-\mathbf{A}_{l_{jk}}\right)\right|^2\right) \tag{34}$$

$$= \left(\frac{1}{D_{\min}^2}\right)^2\frac{1}{2}\left(\sum_i\left|\sum_k(\mathbf{A}_{k_{ik}}-\mathbf{A}_{l_{ik}})\right|^2 m + \sum_j\left|\sum_k\left(\mathbf{A}_{k_{jk}}-\mathbf{A}_{l_{jk}}\right)\right|^2 m\right)$$

Actually, $\sum_i |\sum_k(\mathbf{A}_{k_{ik}} - \mathbf{A}_{l_{ik}})|^2 = \sum_i |\psi|^2$ is the square of degree divergence between $\mathbf{A}_k$ and $\mathbf{A}_l$. Note that

$$\|\mathbf{A}_k - \mathbf{A}_l\|_F^2 = \sum_{i,j}\left(\mathbf{A}_{k_{ij}} - \mathbf{A}_{l_{ij}}\right)^2$$

$$\sum_i |\psi|^2 \leq m\max_i\left(\psi_i\right)^2 \leq m \tag{35}$$

According to Cauchy–Schwarz inequality, we have

$$\left|\sum_k\left(\mathbf{A}_{k_{ik}} - \mathbf{A}_{l_{ik}}\right)\right|^2 = \left[\sum_k(\mathbf{A}_{k_{ik}} - \mathbf{A}_{l_{ik}})\right]^2 \leq m\sum_k(\mathbf{A}_{k_{ik}} - \mathbf{A}_{l_{ik}})^2 \tag{36}$$

Then, it has

$$
\begin{aligned}
\sum_i \left| \sum_k \left( \mathbf{A}_{k_{ik}} - \mathbf{A}_{l_{ik}} \right) \right|^2 &= \sum_i \left[ \sum_k \left( \mathbf{A}_{k_{ik}} - \mathbf{A}_{l_{ik}} \right) \right]^2 \\
&\leq m \sum_i \sum_k \left( \mathbf{A}_{k_{ik}} - \mathbf{A}_{l_{ik}} \right)^2 \\
&= m \left\| \mathbf{A}_k - \mathbf{A}_l \right\|_F^2
\end{aligned}
\tag{37}
$$

Hence, Eq. (34) turns out to be

$$
\begin{aligned}
\left\| \mathbf{T}_2 \right\|_F^2 &\leq \left( \frac{1}{D_{\min}^2} \right)^2 \left[ m^2 \left\| \mathbf{A}_k - \mathbf{A}_l \right\|_F^2 \right] \\
\left\| \mathbf{T}_2 \right\|_F &\leq \frac{m}{D_{\min}^2} \left\| \mathbf{A}_k - \mathbf{A}_l \right\|_F
\end{aligned}
\tag{38}
$$

Thus, Eq. (27) is derived into

$$
\begin{aligned}
\left\| \mathbf{P}_k - \mathbf{P}_l \right\|_F &\leq \left\| \mathbf{T}_1 \right\|_F + \left\| \mathbf{T}_2 \right\|_F \\
&= \frac{1}{D_{\min}} \left\| \mathbf{A}_k - \mathbf{A}_l \right\|_F + \frac{m}{D_{\min}^2} \left\| \mathbf{A}_k - \mathbf{A}_l \right\|_F \\
&= \left( \frac{D_{\min} + m}{D_{\min}^2} \right) \left\| \mathbf{A}_k - \mathbf{A}_l \right\|_F
\end{aligned}
\tag{39}
$$

So far, we can obtain the upper bound of $d_{\mathcal{H}\Delta\mathcal{H}}(\tilde{\mathcal{D}}_k, \tilde{\mathcal{D}}_l)$:

$$
\begin{aligned}
&d_{\mathcal{H}\Delta\mathcal{H}} \left( \tilde{\mathcal{D}}_k, \tilde{\mathcal{D}}_l \right) \\
&\leq 1 + \frac{B_{rank}}{m} \sup_f \left\| f\left( G_k \right) - f\left( G_l \right) \right\|_F \\
&\leq 1 + \frac{B_{rank}}{m} \sup_f B_W^2 B_X \left\| \mathbf{P}_k - \mathbf{P}_l \right\|_F + B_W^2 \left\| \mathbf{X}_k - \mathbf{X}_l \right\|_F \\
&\leq 1 + \sup_f B_W^2 B_X \left( \frac{D_{\min} + m}{D_{\min}^2 m} \right) \left\| \mathbf{A}_k - \mathbf{A}_l \right\|_F + \frac{B_W^2}{m} \left\| \mathbf{X}_k - \mathbf{X}_l \right\|_F \\
&= 1 + \sup_f B_W^2 B_X \left( \frac{1}{D_{\min} m} + \frac{1}{D_{\min}^2} \right) \left\| \mathbf{A}_k - \mathbf{A}_l \right\|_F + \frac{B_W^2}{m} \left\| \mathbf{X}_k - \mathbf{X}_l \right\|_F \\
&\leq 1 + \sup_f B_W^2 B_X \left( \frac{1}{D_{\min} m} + \frac{1}{D_{\min}^2} \right) \left( \left\| \mathbf{A}_k \right\|_F + \left\| \mathbf{A}_l \right\|_F \right) + \frac{B_W^2}{m} \left\| \mathbf{X}_k - \mathbf{X}_l \right\|_F .
\end{aligned}
\tag{40}
$$

It can be seen that $d_{\mathcal{H}\Delta\mathcal{H}}(\tilde{\mathcal{D}}_k, \tilde{\mathcal{D}}_l)$ is bounded by $||\mathbf{A}_k||_F$ and $||\mathbf{A}_l||_F$. When the local causal subgraphs are explored, the edge weights in the adjacency matrices are reduced from 1 to $[0, 1]$, the Frobenious norms of $||\mathbf{A}_k||_F$ and $||\mathbf{A}_l||_F$ are decreased, then the bound of generalization error with respect to the global hypothesis can be shrunk. Hence, the generalization ability of global model learned by the proposed FedATH is enhanced. The proof completes. □

## E. Performance with Increasing Client Number

We verify the performance of different methods as the number of clients increases in Fig. 6. First, the performance of all algorithms degrades as the client number increases, this is because too many clients create fragmentation of information and a disturbance to federated aggregation. Notably, the proposed FedATH consistently maintains the optimal. Second, for different graph datasets, the performance of the algorithms degrades to varying degrees as the client number increases. The reason is that different graph datasets have different levels of importance for the connectivity information, and the loss of connectivity information affects model performance to different degrees.

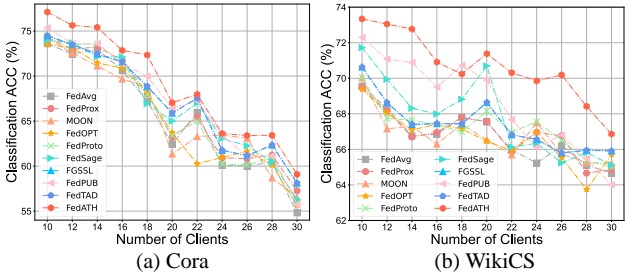

*Figure 6.* The performance for all compared methods as the client number increases.

## F. Convergence Verification

We verify the convergence property of the proposed FedATH compared to other federated learning algorithms in Fig. 7. It can be observed that the performance of FedATH keeps steadily increasing at a faster rate and takes the lead after a certain number of communication rounds. Notably, the convergence of some FGL methods is inferior due to the complex computation process, e.g., FGSSL and FedPUB on PubMed dataset, while the proposed FedATH does not suffer from this case, proving its superiority of convergence.

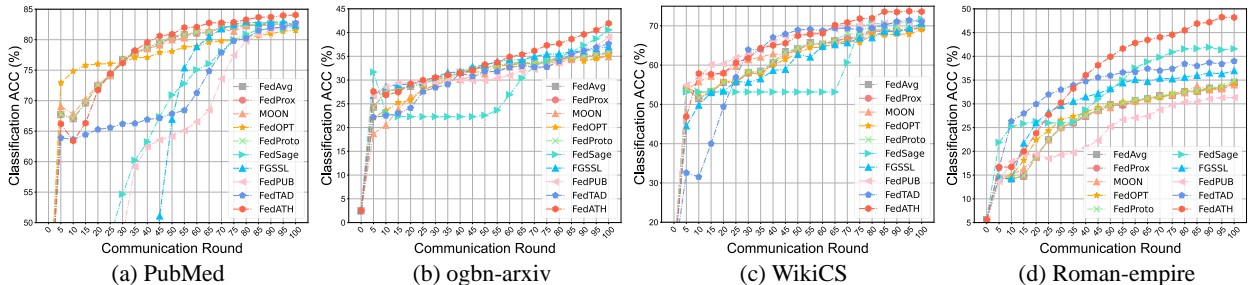

*Figure 7.* The classification ACC curves as the increasing communication round for all compared algorithms, where the client number is set as 10.

