# OpenReview forum: "Less is More: Federated Graph Learning with Alleviating Topology Heterogeneity from A Causal Perspective"
_ICML.cc/2025/Conference — ICML 2025 poster_

### Official Review · Reviewer_eW6P · 2025-03-10

**Overall Recommendation:** 4

**Summary:**

This work proposed a causal subgraph learning method for graph federated learning. The work consists of three critical components. First, the edge evaluator separates the local subgraph into causal subgraph and biased subgraph. Second, the dual-GNN is developed to encode the corresponding subgraphs. Third, the separability of causal and biased subgraphs is enhanced by HSIC. The experimental results on six graph datasets demonstrate the proposed method’s superiority.

**Claims And Evidence:**

Yes, the authors claimed that mining a few of critical causal factors is more effective than introducing new global information, which is reasonable, and experimental results also demonstrated the validity of this argument.

**Essential References Not Discussed:**

Refer to ‘Relation to Broader Scientific Literature’

**Experimental Designs Or Analyses:**

Yes. I have checked the experimental designs and analyses. The authors adopted the Louvain method to split the graph into multiple private subgraphs, and compared the proposed methods with nine typical algorithms. The designs are well thought out. However, the analysis about experimental results needs to be improved, such as hyperparameter study.

**Methods And Evaluation Criteria:**

Yes. This work adheres to the perspective that mining causal subgraphs is more conducive to reducing heterogeneity in graph federated learning, which is a novel viewpoint for this field. Furthermore, the experimental setup and evaluation criteria follow the classic works in graph federated learning, which is reasonable.

**Other Comments Or Suggestions:**

See the weaknesses.

**Other Strengths And Weaknesses:**

Strengths:

   (a) This work proposed a graph decoupling method for graph federated learning, which is significantly different from the existing works. The idea is inspiring, and the writing is good.

   (b) This work demonstrates the effectiveness of the proposed method from two aspects. First, the work conducts a theoretical framework to show the effectiveness. Second, extensive experiments are conducted to prove the claims.

   Weaknesses:

   (a) In Fig.3, the authors claim to capture the causal and biased factors of local subgraphs, but do not explain clearly what a causal factor is and what a biased factor is for local subgraphs.

   (b) The work proposes a generalization analysis for graph federate learning, which has frequently presented in domain adaption and domain generalization. Hence, the differences between them must be explained.

   (c) The computation complexity of the proposed methods must be analyzed.

   (d) The simulation environment should be further introduced in details.

   (e) The font of some figures is too small, it’s suggested to adjust them.

**Questions For Authors:**

See the weaknesses.

**Relation To Broader Scientific Literature:**

This work proposed a decoupled graph representation learning method in the scenario of federated learning, which might be inspired by literature [1]. The proposed Theorem 1 in this work is drawn from literature [2].

[1] Ma J, Cui P, Kuang K, et al. Disentangled graph convolutional networks, ICML, 2019.
[2] Muandet K, Balduzzi D, Schölkopf B. Domain generalization via invariant feature representation, ICML, 2013.

**Theoretical Claims:**

The proposed Theorem 1 shows the upper bound for a graph federated learning system. However, similar generalization analyses have appeared in many domain adaptation and domain generalization efforts, so it is difficult to see a difference between them.

---

> ### Author Rebuttal · Authors · 2025-03-31
>
> Many thanks for your valuable feedback, we carefully reply your concerns as follows.
>
> **Response to Weakness(a):** The topology of a graph has a significant influence on node embedding. However, inspired by causal learning, we argue that only the critical topological information is a direct determinant for the class of node representations, which is called the causal factor, while the remaining topological information is called the biased factor.
>
> **Response to Weakness(b):** Admittedly, the general form of the algorithm's generalizability analysis has its roots in domain generalization theory. Indeed, most studies on domain adaptation and domain generalization further analyze the distribution discrepancy $d_{\mathcal{H}\Delta \mathcal{H}}(\tilde{\mathcal{D}}_s,\tilde{\mathcal{D}}_t)$.
> Similarly, we analyze the term for the upper bound, but for graph data, it is necessary to consider not only differences in feature distributions, but also differences between topological distributions. Therefore, we perform a customized upper bound analysis, which is not available in the existing work on federated graph learning.
>
> **Response to Weakness(c):** The computational complexity of FedATH mainly comes from the computation process on the clients and consists of three components. Let us take the computation of a client as an example. First, the forward propagation of dual-GCN costs $O(|E|L)$, where $|E|$ denotes the number of edge and $L$ denotes the dimension of embedding. Second, the computation of edge evaluator costs $O(|E|d)$, where $d$ denotes the feature dimension of nodes. Third, the calculation of HSIC criterion takes $O(N^3)$. Overall, the computational complexity of the proposed FedATH is $O(|E|L+|E|d+ N^3)$.
>
> **Response to Weakness(d):** We use Python and PyTorch to conduct the simulation environment, and perform the experiments of FedATH and compared experiments on a server equipped with an Intel(R) Core(TM) i9-10980XE processor, an RTX 3090 GPU, and 128 GB of RAM. Adam is selected as the optimizer, and the learning rate is set as 0.001. The numbers of communication rounds and local training epochs are fixed as 100 and 3, respectively.
>
> **Response to Weakness(e):** We will optimize the figures in the following revision for a better presentation.
>
> **Response to Essential References Not Discussed:**
> Study [1] proposes to decouple graph data into features of multiple potential subspaces through a new neighborhood routing mechanism, which mainly aims to mine the decoupling factors behind the data and enhance the interpretability and generalization of the model. However, this mechanism may not be suitable for solving topological heterogeneity problems in federated graph learning, since the separation of multiple decoupling factors is computationally and communication-unfriendly, and it is difficult to judge which factor is really needed to be shared by all clients. In contrast, the proposed FedATH decouples the node representations from the topology level and only shares the causal networks, which is computationally and communicatively acceptable.
> Study [2] proposes a kernel-based optimization method to reduce the discrepancy between the source and target domains and proves it theoretically. However, as described in Response to W(b), for federated graph learning, the effect of topology on the generalization error needs to be taken into account, and thus we tailor an analytical approach that is not found in the exsiting studies.
>
> [1] Ma, J., Cui, P., Kuang, K., Wang, X., Zhu, W. Disentangled Graph Convolutional Networks. ICML, pp. 4212-4221, 2019.
>
> [2] Muandet, K., Balduzzi, D., Schölkopf, B. Domain generalization via invariant feature representation. ICML, pp. 10-18, 2013.

---

### Official Review · Reviewer_eaWv · 2025-03-11

**Overall Recommendation:** 4

**Summary:**

To address the topology heterogeneity of FGL, the authors proposed an interesting idea, namely, Less is More. Concretely, the unnecessary edges are discarded while the necessary edges are maintained. The CE loss and NE loss are separately used to train the corresponding GNNs. The HSIC loss is adopted to enforce the training of the dual-GNN. From both theorical and experimental perspectives, the authors verify the superiority of the proposed FedATH.


## update after rebuttal
After the Reviewer-author discussion phase, I maintain my score and explicitly support acceptance.

**Claims And Evidence:**

Yes

**Essential References Not Discussed:**

N/A

**Experimental Designs Or Analyses:**

In order to verify the effectiveness of the proposed method, the authors conducted a large number of experiments, and the experimental design and analysis are fully considered.

**Methods And Evaluation Criteria:**

Yes, the traditional FGL methods tend to create an extra global information for correct the local training, while the presented idea goes the other way and advocate discarding the unnecessary edges.

**Other Comments Or Suggestions:**

None.

**Other Strengths And Weaknesses:**

Strength:
    First, this paper proposes a new idea that discarding unimportant edges is more conducive to alleviating topological heterogeneity in federated graphs, which is different from the existing works. Second, many experiments are conducted on real-word graph datasets, the results demonstrate that the proposed FedATH outperforms other methods.

Major Weaknesses:
    1) The proposed FedATH is compared with nine FL algorithms, only four of which are tailored to the distributed graphs. At present, there are many FGL studies. Therefore, it’s suggested to compare FedATH with more FGL methods.
    2) The authors split a complete graph into multiple subgraphs via Louvain algorithm. When the training of FL finishes, how do the author test the algorithms? Are they performed on the original graph? The details of test period should be explained.

 Minor Weaknesses:
1)	There are too many symbols in the paper, a Table is needed for summarizing them.
2)	Some typos and grammatical errors still appear, the authors should carefully check the paper.
3)	The format for several figures is required to be changed, especially for the text size.

**Questions For Authors:**

None.

**Relation To Broader Scientific Literature:**

The proposed FedATH aims to reduce the topology heterogeneity of FGL [1,2,3], discarding the unnecessary edges while maintaining the necessary edges. Compared to the existing literature, the proposed FedATH is well motivated and can make sense of the FGL field.
[1] Chen F, Li P, Miyazaki T, et al. Fedgraph: Federated graph learning with intelligent sampling. IEEE Transactions on Parallel and Distributed Systems, 2021, 33(8): 1775-1786.
[2] Xie H, Ma J, Xiong L, et al. Federated graph classification over non-iid graphs. Advances in neural information processing systems, 2021, 34: 18839-18852.
[3] Tan Y, Liu Y, Long G, et al. Federated learning on non-iid graphs via structural knowledge sharing, Proceedings of the AAAI conference on artificial intelligence. 2023, 37(8): 9953-9961.

**Theoretical Claims:**

Theorem 5.1 and Corollary 5.2 give the generalization bound for FGL, the proof is complete, which well supports the soundness of the proposed FedATH.

---

> ### Author Rebuttal · Authors · 2025-03-31
>
> Thank you very much for your valuable comments, the detailed responses are provided as follows.
>
> **Response to Weakness1:** We have collected two new federated graph learning methods FGGP [1], FedGTA [2], and compared them with the proposed FedATH. The experimental results are reported in the following table, it can be observed that the proposed FedATH is more superior than the new FGL methods.
>
> | Method   | Cora (K=10) | Cora (K=15) | Cora (K=20) | Photo (K=10) | Photo (K=15) | Photo (K=20) |
> |----------|-------------|-------------|-------------|--------------|--------------|--------------|
> | FGGP [1] | 73.96       | 71.75       | 65.84       | 88.56        | 86.27        | 84.25        |
> | FedGTA [2] | 75.36     | 71.43       | 65.87       | 88.28        | 87.02        | 85.17        |
> | FedATH   | **77.90**       | **73.42**       | **67.97**       | **90.33**        | **88.61**        | **85.50**        |
>
> [1] Wan, G., Huang W., Ye M. Federated graph learning under domain shift with generalizable prototypes. AAAI, pp. 15429-15437, 2024.
>
> [2] Li, X., Wu, Z., Zhang W., etc. FedGTA: topology-aware averaging for federated graph learning. VLDB, pp. 41-50, 2024.
>
> **Response to Weakness2:** In fact, when we divide the subgraphs to multiple clients, each client has its own training set and test set. Thus, after federated training, the global model is tested on each client and eventually the average performance of all clients is reported.
>
> **Response to Weakness3:** We summarize the meanings of main symbols in the following table.
> | Symbol | Meaning |
> |----------------------|------------------------------------------------------------------------------------------------------------------|
> | $G=(V, E, X)$ | A graph dataset, node set, edge set, feature matrix |
> | $G_k^c$, $G_k^b$ | The causal subgraph, the biased subgraph |
> | $A_k$, $H_k$ | The adjacency matrix, the latent embedding for the k-th client |
> | $C$, $K$ | The number of classes, the number of clients |
> | $N_k$, $N$ | The number of nodes for the k-th client, the number of total nodes |
> | $c_{ij}$, $w_{ij}$ | The importance score for edge $e_{ij}$, the normalized score |
> | $\Omega_k^c$, $\Omega_k^b$ | The causal edge mask, the biased edge mask for the k-th client |
> | $f_k^c$, $f_k^b$, $\Phi_k$ | The causal GCN, the biased GCN, the edge evaluator for the k-th client |
> | $\lambda$ | The trade-off parameter |
>
> **Response to Minor Weaknesses:** Thanks for the meticulous advice. We will improve the presentation in a revision version.

---

> > ### Comment · Reviewer_eaWv · 2025-04-03
> >
> > Thanks to the authors for their response. I notice that the authors test the effect of the global model on the local test set at the end of the local training, while some algorithms are tested for performance before the local training starts, and different testing methods may also lead to differences in the results. The authors should compare the different testing methods.

---

> > > ### Author Response · Authors · 2025-04-05
> > >
> > > Thank you for this valuable comment. We test the performance at the beginning of a new communication round on Cora and Photo datasets, the ACC results are reported in the following table. It can be seen that the proposed FedATH still achieves the superior performance compared to the other FL and FGL methods.
> > >
> > > | Method     | Cora K=10 | Cora K=15 | Cora K=20 | Photo K=10 | Photo K=15 | Photo K=20 |
> > > |------------|-----------|-----------|-----------|------------|------------|------------|
> > > | FedAvg     | 75.78     | 70.88     | 63.09     | 86.82      | 68.86      | 77.00      |
> > > | FedProx    | 76.49     | 71.74     | 64.75     | 87.34      | 68.70      | 77.06      |
> > > | MOON       | 77.02     | 71.58     | 62.50     | 86.40      | 71.43      | 76.05      |
> > > | FedOPT     | 74.91     | 67.63     | 60.35     | 86.25      | 72.12      | 75.81      |
> > > | FedProto   | 75.43     | 70.46     | 63.94     | 88.31      | 71.34      | 78.40      |
> > > | FedSage+   | 76.67     | 72.64     | 65.29     | 87.35      | 74.00      | 78.61      |
> > > | FGSSL      | 76.94     | 73.64     | 64.87     | 82.54      | 74.15      | 77.43      |
> > > | FedPUB     | 76.78     | 72.17     | 66.98     | 87.54      | 72.30      | 79.60      |
> > > | FedTAD     | 77.38     | 71.84     | 66.47     | **89.66**  | 74.11      | 78.37      |
> > > | FedATH     | **77.55** | **73.78** | **68.54** | 89.33      | **75.66**  | **79.82**  |

---

### Official Review · Reviewer_oUdB · 2025-03-11

**Overall Recommendation:** 4

**Summary:**

This paper proposed a reduced-edge based federated graph learning method that aims to mitigate the effects of topological heterogeneity on federated learning. Specifically, the proposed FedATH assesses the importance of each edge via an edge evaluator. Thus, the local subgraph is divided into causal and biased subgraphs. The causal subgraph is deemed to capture the essential information and perform the classification.

**Claims And Evidence:**

The claims of this paper are clear, and the experimental results support the claims.

**Essential References Not Discussed:**

I'm not aware of any additional literature that needs to be discussed.

**Experimental Designs Or Analyses:**

Overall, the experiment parts are complete, but I still have some concerns:

1.	More analysis should be explained about the effects of sharing different components.

2.	Large-scale clients are an important challenge for federated learning, and I would like to know how the proposed algorithm performs in the face of large-scale clients.

**Methods And Evaluation Criteria:**

Yes, the proposed methods and evaluation criteria make sense to the federated graph learning field.

**Other Comments Or Suggestions:**

N/A

**Other Strengths And Weaknesses:**

Strengths:

1.	In terms of topological heterogeneity in federated graph learning, the authors provide new ideas for solving the problem by reducing data information rather than introducing new information.

2.	It is remarkable that this paper theoretically guarantees that the proposed method can effectively solve the topological heterogeneity problem.

3.	This paper is well-written and the presentation is excellent.

Weaknesses:

1.	The authors mention dividing this map into causal and biased subgraphs, however, their exact definitions are not made clear.

2.	In fact, I don't fully understand what the authors mean by topological heterogeneity and how it differs from the common data heterogeneity of federal learning.

3.	The questions about the experiments refer to the above ‘Experimental Designs and Analysis’.

**Questions For Authors:**

In addition to the above weaknesses, I still have one more concern:

To decouple the causal graph and the biased graph, the authors encoded the two graphs locally using two GCNs. This process seems to bring higher computational cost, whether the proposed method consumes a much higher runtime than the comparison method?

**Relation To Broader Scientific Literature:**

This paper follows the problem setting of literature [1] and [2] and aims to address the difficulties posed by topological heterogeneity for federated learning. However, unlike them, this work mitigates the problem of topological heterogeneity by reducing data information rather than increasing it. Furthermore, the proposed method adopts the HSIC criteria [3].

  [1] Ke Zhang, Carl Yang, Xiaoxiao Li, Lichao Sun, and SiuMing Yiu. Subgraph federated learning with missing neighbor generation.

  [2] Yinlin Zhu, Xunkai Li, Zhengyu Wu, Di Wu, Miao Hu, and Rong-Hua Li. Fedtad: Topology-aware data-free knowledge distillation for subgraph federated learning.

  [3] Greenfeld, Daniel and Shalit, Uri. Robust learning with the hilbert-schmidt independence criterion.

**Theoretical Claims:**

I check the Theorem 5.1 and Corollary 5.2, I believe that they are correct.

---

> ### Author Rebuttal · Authors · 2025-03-30
>
> Thank you for your professional advices, the detailed responses are written as follows.
>
> **Response to Weakness1:** The difference between causal and biased graphs is whether they contain critical edge information. We use an edge evaluator to assess the importance of each edge. Subgraphs that contain information about important edges are called causal subgraphs, and subgraphs that contain information about the remaining edges are called biased subgraphs.
>
> **Response to Weakness2:** In traditional federated learning, the samples are isolated from each other and do not affect each other in model training, the model is only affected by differences in data label distribution or feature distribution. In contrast, for federated graph learning, the samples have connecting edges between them, and the sample representations are affected by connecting edges between different nodes. For example, the same class of nodes may connect different classes of nodes in different clients, then the representations minght be heterogeneous even for same classes of nodes in various clients. Therefore, topological heterogeneity is different from traditional label heterogeneity or feature heterogeneity.
>
> **Response to Weakness3:** First, to verify that sharing only causal networks is sufficient, we tested the algorithm performance when sharing different network components. In most scenarios, sharing only the causal network can achieve optimal results, which indicates that the causal factor is a key component for affecting the performance of the model. It should also be noted that sharing edge evaluators may also be beneficial to improve the performance, due to the fact that the fusion of edge information from different clients facilitates the quality of the evaluators. Second, the experimental results on Photo dataset with large-scale clients in the following table. We can see that the proposed FedATH still takes the leading position in the scenario of large-scale clients.
>
> | Method    | Photo (K=50) | Photo (K=80) | Photo (K=100) |
> |-----------|--------------|--------------|---------------|
> | FedAvg    | 84.20        | 83.11        | 82.38         |
> | FedProx   | 84.23        | 83.11        | 82.41         |
> | MOON      | 84.79        | 82.99        | 82.37         |
> | FedOPT    | 80.56        | 78.78        | 79.16         |
> | FedProto  | 75.04        | 74.16        | 72.20         |
> | FedSage+  | 84.88        | 83.44        | 82.98         |
> | FGSSL     | 85.71        | 83.45        | 83.38         |
> | FedPUB    | 83.69        | 82.18        | 79.75         |
> | FedTAD    | 84.97        | 83.72        | 83.23         |
> | FedATH    | **86.10**        | **84.16**        | **83.43**         |

---

> > ### Comment · Reviewer_oUdB · 2025-04-03
> >
> > The author addressed most of my questions, but I still have one concern, which is posted in “Questions for Authors”.

---

> > > ### Author Response · Authors · 2025-04-05
> > >
> > > Thanks for your insightful comment. We compare the running time (second) on Cora and Photo datasets for all compared methods. Admittedly, the running time of the proposed FedATH is increased due to the dual GCN coding structure and the computation of the HSIC loss, but it is well within acceptable limits. Especially on the Photo dataset, the running efficiency of our algorithm is better than the FedProto, FedSage+, and FedTAD.
> > >
> > > | Method     | Cora K=10 | Cora K=15 | Cora K=20 | Photo K=10 | Photo K=15 | Photo K=20 |
> > > |------------|-----------|-----------|-----------|------------|------------|------------|
> > > | FedAvg     | 12.36     | 18.08     | 23.63     | 12.23      | 18.89      | 23.36      |
> > > | FedProx    | 15.01     | 22.46     | 30.72     | 15.28      | 22.73      | 30.36      |
> > > | MOON       | 21.76     | 33.19     | 43.56     | 21.77      | 32.15      | 46.17      |
> > > | FedOPT     | 12.73     | 18.06     | 23.95     | 12.91      | 18.76      | 24.39      |
> > > | FedProto   | 49.74     | 56.38     | 62.68     | 112.01     | 125.97     | 131.15     |
> > > | FedSage+   | 20.02     | 32.03     | 42.86     | 78.27      | 118.02     | 142.35     |
> > > | FGSSL      | 28.13     | 41.85     | 56.09     | 30.68      | 42.65      | 57.16      |
> > > | FedPUB     | 16.95     | 26.10     | 35.71     | 17.63      | 25.99      | 36.12      |
> > > | FedTAD     | 535.85    | 890.41    | 1315.30   | 679.24     | 1027.08    | 1513.65    |
> > > | FedATH     | 51.03     | 75.41     | 86.42     | 55.91      | 79.57      | 104.40     |

---

### Official Review · Reviewer_AFBr · 2025-03-13

**Overall Recommendation:** 3

**Summary:**

The paper proposes to divide the local graph into causal subgraph and biased subgraph  for alleviating the topology heterogeneity, the causal subgraph possesses the key information for the downstream task, and the biased subgraph possesses the confusing information. Thus, only the causal graph neural networks are shared in the server. The generalization bound for the proposed FedATH is derived, and the experimental results also support the authors’ argument.

**Claims And Evidence:**

The authors claim that less edges are more conducive to handling the topology heterogeneity, then the generalization analysis and experimental results support their claims.

**Essential References Not Discussed:**

No, the current literature discussion is adequate.

**Experimental Designs Or Analyses:**

Overall, the experimental section is ok.

**Methods And Evaluation Criteria:**

The proposed FedATH explores the local causal subgraph and improves the performance on the test datasets, making sensing of the domain of federated graph learning.

**Other Comments Or Suggestions:**

See above weakness.

**Other Strengths And Weaknesses:**

Strengths:

1. In Figure 1, the authors visualize what topology heterogeneity is and the shortcomings of existing methods, showing that the proposed FedaATH is well motivated.

2. The authors provide a rigorous derivation of the generalizability analysis for the proposed FedATH, and the conclusion shows that mining causal subgraphs and abandoning biased subgraphs is beneficial to enhance the generalizability of the model.

Weaknesses:

1. Corollary 5.2 illustrates that mining the causal subgraphs are beneficial for reducing the topology heterogeneity. Consider an extreme case when all edges are removed, whether the performance is optimal.

2. In the experiments, the FL algorithms are tested with up to 20 clients. However, more clients will be involved in the practical application. So, it’s necessary to conduct the experiments with more clients, at least 100 clients.

3. In related work, only FL and FGL are introduced. The causal learning is a key technique, a review about it is indispensable.

**Questions For Authors:**

1. The code of the proposed FedATH is unavailable, to validate its reproducibility, it’s suggested to release the code.

2. Whether the so-called topological heterogeneity is just a form of label heterogeneity on a graph, and what is the essential difference between the two.

3. In Eq. (9), is it reasonable to seek for an expectation for all nodes? Is it more appropriate to seek for an expectation for nodes that are only labeled?

**Relation To Broader Scientific Literature:**

The representation division [1][2] has been applied into some scenarios, it aims at capturing an invariant representation from the coupled representation. The proposed FedATH is also based on this theory.

[1] On learning invariant representations for domain adaptation. In ICML 2019.
[2] Self-supervised learning of pretext-invariant representations. In CVPR 2020.

**Theoretical Claims:**

I checked the generalization analysis in Section 5 and the proof process in Appendix, I’m leaning towards them being right.

---

> ### Author Rebuttal · Authors · 2025-03-30
>
> Thank you very much for your valuable comments, we response your concerns as follows.
>
> **Response to Weakness1:** We test the performance of several federated learning methods when all the edges are removed on Cora dataset. It can be seen that although topological heterogeneity causes a decline in federated learning, the topology still favors feature learning.
> Therefore, it is not the case that removing the topology guarantees that optimal performance will be achieved. We give an upper bound on the generalization error in the theorem. However, for practical purposes, a learning system has a lower bound on its generalization error, but the analysis of the lower bound is difficult to infer.
>
> | Method    | Cora (K=10) | Cora (K=15) | Cora (K=20) |
> |-----------|-------------|-------------|-------------|
> | FedAvg    | 62.57       | 64.99       | 61.26       |
> | FedProx   | 64.22       | 65.43       | 62.47       |
> | MOON      | 63.97       | 65.34       | 61.60       |
> | FedOPT    | 62.75       | 64.61       | 62.17       |
> | FedProto  | 60.72       | 62.21       | 60.17       |
>
> **Response to Weakness2:** Validating the adaptability of federated learning algorithms in large-scale client scenarios is necessary. We conduct the experiments with 50, 80, and 100 clients on Photo dataset. It can been seen that the proposed FedATH still outperforms other methods in the scenario of large-scale clients, demonstrating that it has good adaptability in face of more clients.
>
> | Method    | Photo (K=50) | Photo (K=80) | Photo (K=100) |
> |-----------|--------------|--------------|---------------|
> | FedAvg    | 84.20        | 83.11        | 82.38         |
> | FedProx   | 84.23        | 83.11        | 82.41         |
> | MOON      | 84.79        | 82.99        | 82.37         |
> | FedOPT    | 80.56        | 78.78        | 79.16         |
> | FedProto  | 75.04        | 74.16        | 72.20         |
> | FedSage+  | 84.88        | 83.44        | 82.98         |
> | FGSSL     | 85.71        | 83.45        | 83.38         |
> | FedPUB    | 83.69        | 82.18        | 79.75         |
> | FedTAD    | 84.97        | 83.72        | 83.23         |
> | FedATH    | **86.10**        | **84.16**        | **83.43**         |
>
> **Response to Weakness3:** We have noticed that a review of causal learning is necessary, a review about causal learning is written as follows.
>
> Causal learning [1,2] aims to uncover causal factors that affects the data distributions and has gained widespread attention in machine learning.
> Mining causal factors is often used in invariant representation learning. For example, Lu et al. [3] aimed to discover the independent variables in the data under the nonlinear setting.
> [4] further disentangled the causal assumptions into invariant factors and nonstable factors.
> [5] proposed a causal variational autoencoder that captured the causal structures in the data.
> In recent years, to learn invariant representations in graph learning, some studies have also absorbed the idea of ​​causal learning.
> For instance, [6] explored a key subgraph from the initial graph to capture the most important information.
> [7] proposed a graph contrastive invariant learning to enhance the performance of graph representation.
> [8] introduced the graph autoencoder to learn the invariant graph representations across multiple domains.
> Inspired by the causal learning, we believe that the local graph consists of causal subgraph and biased subgraph, and sharing the causal information in them help enhance the generalization of the global model.
>
> [1] Pearl, J. Causality. 2009.
>
> [2] Causality for trustworthy artificial intelligence: status, challenges and perspectives. ACM Computing Surveys,  2025.
>
> [3] Invariant causal representation learning for out-of-distribution generalization. ICLR, 2022.
>
> [4] Invariant and transportable representations for anti-causal domain shifts. NIPS, 2022.
>
> [5] Causalvae: Disentangled representation learning via neural structural causal models. CVPR,  2021.
>
> [6] Learning causally invariant representations for out-of-distribution generalization on graphs. NIPS, 2022.
>
> [7]  Graph contrastive invariant learning from the causal perspective. AAAI, 2024.
>
> [8] Learning causal representations based on a gae embedded autoencoder. IEEE TKDE, 2025.
>
> **Response to Question1:**  We have released the code at https://anonymous.4open.science/r/FedATH-9517.
>
> **Response to Question2:** Label heterogeneity emphasizes that the label distribution for independent samples creates a bias. Topological heterogeneity, on the other hand, emphasizes more on the connecting edge relationship between nodes to produce variance. In other words, label heterogeneity can lead to topological heterogeneity, while topological heterogeneity may not be caused by label heterogeneity.
>
> **Response to Question3:** The purpose of Eq. 9 is to smooth the predictive distribution output by the biased network, not just for labeled nodes, but for all nodes.

---

> > ### Comment · Reviewer_AFBr · 2025-04-03
> >
> > The authors have addressed my questions. I maintain my score and support its acceptance.

---

### Decision · Program_Chairs · 2025-05-01

**Decision:**

Accept (poster)

**Comment:**

In this submission, the authors propose a new federated graph learning method, which aims to tackle the challenge of topology heterogeneity, different from the traditional data heterogeneity. The proposed method is designed from a causal perspective, and extensive experiments on several tasks also confirm its benefits and great potentials.

During the rebuttal, all the reviewers appreciate this work. Please incorporate the new important discussion and experiments into the final version. Hope the authors find the discussions with reviewers useful and make this submission a better one.